# A pictural guide to postmortem examination of elephants

Almuth Falkenau[1]*, Ninja Kolb[1], Alexandra Rieger[1], Isabelle Lutzmann[1¤a], Katharina Erber[1], Clara Kaufhold[1¤b], Lina Eddicks[1], Marco Rosati[1¤c], Sonja Fiedler[1], Anna Gager[1¤d], Effrosyni Michelakaki[1], Elena Dell'Era[1], Timo Lorenzen[1], Martin Zöllner[2], Andreas Brühschwein[2], Andrea Meyer-Lindenberg[2], Julia Heckmann[3], Marco Roller[3], Lukas Reese[3], Barbara Lang[3], Markus Menzinger[4], Nicole Richter[4], Robert Fitz[4], Lukas Pfaudler[4], Christine Lendl[5], Hanspeter W. Steinmetz[6], Christine Gohl[6], Monir Majzoub-Altweck[1], Kaspar Matiasek[1], Andreas Blutke[1]

1 Institute of Veterinary Pathology, Center for Clinical Veterinary Medicine, Ludwig-Maximilians-Universität München, Munich, Germany, 2 Clinic of Small Animal Surgery and Reproduction, Center for Clinical Veterinary Medicine, Ludwig-Maximilians-Universität München, Munich, Germany, 3 Zoologischer Stadtgarten Karlsruhe, Karlsruhe, Germany, 4 Tierärztliche Klinik Gessertshausen Altano GmbH, Gessertshausen, Germany, 5 Tiergesundheitszentrum München, Munich, Germany, 6 Münchner Tierpark Hellabrunn AG, Munich, Germany

¤a Current Address: Landratsamt Bad Tölz, Bad Tölz, Germany.
¤b Current Address: Metabolic Biochemistry, Biomedical Center (BMC), Faculty of Medicine, Ludwig-Maximilians-Universität München, Munich, Germany.
¤c Current Address: AstraZeneca Computational Pathology GmbH, Munich, Germany.
¤d Current Address: Department for Pathology, Parasitology and Bee Diseases, Bavarian Health and Food Safety Authority, Oberschleissheim, Germany.
* falkenau@patho.vetmed.uni-muenchen.de

## Abstract

The necropsy of an elephant represents a rare event for most veterinary pathology facilities outside of Africa and Asia. Here, we report a comprehensible, abundantly illustrated, step-by-step protocol adapted to the special technical and anatomical peculiarities of elephant necropsies with regard to the needed equipment, aspects of transmissible disease prevention and workplace safety, personnel, time efforts, as well as important elephant diseases and their zoonotic potential. Detailed instructions for dissection, macroscopic examination and sampling of all relevant organs and tissues are provided, along with checklists for preparation and smooth execution of elephant necropsies. Using the featured protocols, a complete elephant necropsy with generation of extensive samples for histological, microbiological, and molecular analyses can be performed by 10–12 persons (5–6 pathologists and 5–6 assistants) within 4–6 hours in case of an adult animal.

## Introduction

The necropsy of an elephant is a rare event in most veterinary pathology institutions in western industrialized countries. In the author's Institute of Veterinary Pathology of the

**Data availability statement:** All relevant data are included in the manuscript and its Supporting Information files. The links for Supporting Video Files , S1 and S2 can be found at the following links, respectively: https://syncandshare.lrz.de/dl/fiWGfUJtwCpy-D8NVGj1Qh8/S1_Video%20Extraction%20of%20the%20elephant%20brain.mp4, https://syncandshare.lrz.de/dl/fiTAzAyoCTETgdbo6f-pD45/S2_Video_LMU%20Protocol%20for%20Systematic%20Dissection%20of%20an%20Elephant%20Brain.mp4'.

**Funding:** The author(s) received no specific funding for this work.

**Competing interests:** The authors have declared that no competing interests exist.

LMU Munich, *e.g.,* 19 elephants were necropsied between 1994 and 2022 (*i.e.,* < 1 case/year). Most elephants (Asian elephants, *Elephas maximus*, EM, and African elephant, *Loxodonta africana*, LA) are kept in zoological institutions and in circuses outside their home range countries. In Germany, for example, currently (status as of 2025) 126 elephants (99 cows and 27 bulls) are kept in 27 zoos, and 5 (cows only) in 2 circuses [1,2]. In the unfavorable case of death, a postmortem examination is obligatory for elephants in all scientific guided institutions. On top of this, the fact elephant death under human care is regularly accompanied by a lively public and media interest. Necropsy promotes clarification and a high degree of transparency in public communication. The dissection of an elephant also provides a rare opportunity to collect organ and tissue samples for ongoing international studies examining elephant diseases and may also awaken the interdisciplinary interest of other scientists, requesting for collection of special organ/tissue specimens. Elephants kept under human care imply an important potential for scientific work in this species, as benefit for the worldwide population including the remaining animals in their natural habitats. The sheer size of these animals, as well as distinct anatomical peculiarities, present specific challenges to the pathologist(s) performing the necropsy. These include infrastructural requirements, such as appropriate facilities, personnel, and special equipment, as well as particular aspects of personal infection prophylaxis and occupational safety. Thus, the planning and execution of a successful elephant necropsy represents a challenging and responsible task, to receive the maximal gain of knowledge. So far, some published protocols are available, providing helpful general advice for elephant necropsies, but mostly in text form [3–7]. The present work features abundantly illustrated, detailed, step-by-step protocols guiding through the dissection process, with particular attention to those parts of the necropsy that might pose a problem for pathologists not experienced in elephant necropsies, *e.g.*, the extraction of the brain from the skull. The protocols include recommendations for the organization of the necropsy, such as handling of the elephant body upon delivery, required personnel and time, instruments and supplies, important personal infection prophylaxis and workplace safety arrangements, sampling schemes for organs with an elephant-specific anatomy, field-tested checklists, and the expected amount of condemned animal material for disposal. It is supplemented by brief summaries of the most relevant aspects of elephant anatomy and physiology, as well as of common and specific elephant diseases under human care and their zoonotic potential.

## Contents

This elephant necropsy guide includes important information regarding forethoughts, practical prearrangements and briefing of the entire necropsy team (**Section 1–5**) as well as an abundantly illustrated step-by-step protocol of elephant body dissection (**Section 6**). **Table 1** outlines the different subjects.

## 1 Preliminary remarks

The essential sequence of examinations during the necropsy of an elephant does not significantly differ from a postmortem in any other common mammal species, including the anamnesis, external adspection, dissection and gross examination of

**Table 1. Section overview.**

| Section | Subject |
|---------|---------|
| 1 | Preliminary remarks |
| 2 | Elephant anatomy and physiology |
| 3 | Common and specific diseases of elephants under human care |
| 4 | Workplace safety/infection prophylaxis and personnel protective equipment (PPE) |
| 5 | Prearrangements for elephant necropsy |
| 5.1 | Necropsy site |
| 5.2 | Materials and equipment |
| 5.3 | Personnel |
| 5.4 | Dissection notes/documentation of findings and organ/tissue sample lists |
| 5.5 | Samples for superordinate (inter)national elephant studies |
| 6 | Postmortem procedure |
| 6.1 | Clinical history |
| 6.2 | Storage, transport and unloading of an elephant body |
| 6.3 | External examination |
| 6.4 | Dismemberment of the elephant body |
| 6.5 | Dissection of distinct organs/tissues |
| 6.5.1 | Dissection of the head |
| 6.5.2 | Musculoskeletal system, spinal cord, peripheral nerves, and feet |
| 6.5.3 | Spleen and gastrointestinal system |
| 6.5.4 | Respiratory and cardiovascular system |
| 6.5.5 | Urogenital system |
| 6.5.6 | Endocrine system |
| 6.6 | Sampling of organ/tissue specimens |
| 6.7 | Disposal of the carcass and disinfection |
| 6.8 | Time requirements |
| 6.9 | Animals, ethics statement |

body cavities, organs and locomotor system, collection of tissue samples, documentation and reporting of findings, and disposal of animal waste. The relevant peculiarities of elephant necropsies are caused by the required number of personnel (**Section 5.3**), as well as by some distinctive features of elephant anatomy (**Section 2**), and necessary precautions against zoonotic elephant diseases, such as tuberculosis (TB) (**Section 3**), but, above all, by the unusually large size of the corpse and the organs/tissues. The latter implies the handling and transport of heavy body parts and the use of "heavy equipment" and instruments (**Section 5.2**), which is associated with an increased hazard potential. Therefore, safety precautions are followed, and adequate personnel protective equipment is worn by all participants of an elephant necropsy (**Section 4**). An elephant necropsy is ideally led by a single, experienced pathologist in charge, who plans, organizes, delegates and controls the necessary prearrangements (**Section 5**), the division of the staff into separate necropsy groups (**Section 5.3**), the process of the necropsy (**Section 6**), the monitoring of the compliance with the set safety regulations, the collection of organ/tissue samples (**Section 6.6**), their processing and shipment (if appropriate), the documentation and reporting of gross findings (**Section 5.4**), and the adequate disposal of the carcass and final disinfection (**Section 6.7**).

## 2 Elephant anatomy and physiology

Key data on the general body dimensions, life expectancy and reproduction of elephants, as well as anatomical peculiarities are summarized in **Table 2**. If appropriate, additional information is provided in the respective sections of the necropsy

**Table 2. Key data of elephant physiology and anatomy.**

| Parameter | Asian elephant (*Elaphas maximus*, EM) | African elephant (*Loxodonta africana*, LA) |
|---|---|---|
| **Shoulder height (m)** | 2.6-2.9 (male) 2.3-2.5 (female) | 3.0-3.3 (male) 2.5-2.7 (female) |
| **Body weight (t)** | 3.7-4.5 (male) 2.3-3.7 (female) | 4.1-5.0 (male)2.3–4.0 (female) [5,8,9] |
| **Age (years)** | 60-70 [8] | |
| **Gestation (years)** | 2 (interbirth interval: 4–5; weaning: ~3) [8] | |
| **Birth weight (kg)/ size (cm)** | 120/ 85 [8] | |
| **Sexual maturity (years)** | 14-15 (male) 9 (female) [8] | |
| **Adulthood (years)** | 18 [8] | |
| **Body temperature (°C)** | ~36 [8] | |
| **Heart rate (bpm)** | ~30 (increased when elephant is lying down) [8,10] | |
| **Skull** | Skull skeleton contains extensive, honeycomb-like sinuses [8,11] for weight reduction (**Figs 11D** and **12L**). | |
| **Eyes** | No lacrimal apparatus. Moisturization by Harderian glands. Robust nictitating membrane [11]. | |
| **Brain** | Weight: 3.5–5.5 kg. Gyrated, prominent temporal lobes [8,11], (**Fig 12K**). | |
| **Auditory sense** | Capable of detection of very low frequencies (1 kHz) [12]. | |
| **Temporal gland** | Unique in elephants. Secretion of a characteristic fluid during the mating season ("*musth*"), associated with sexual behavior, particularly in male elephants [8,11] | |
| **Trunk** | Essential for ingestion of food and water and used for snorkeling when swimming. Complex musculature; Continuous nasal septum, cartilage at the base of the nostrils; Innervation by the proboscis nerve lateral at the trunk (maxillary and facial nerve branches); Two (LA), respectively one (EM) grasping fingers at the trunk tip [8,11] (**Fig 2C**). | |
| **Teeth** 1.0.3.3 0.0.3.3 [dentition formula] | **Tusks**: Second dentition at 6–12 months; Continuous growth **Pre/molars**: Polyphyodont, *i.e.,* 1 pre/molar is in use at a time. Abraded chewing teeth fall out and are replaced 5 times by new, forward-moving tooth formed in the back of the jaw at approximately 2–3 years, 6 years, 9–15 years, and <40 years of age. The (final) sixth set of teeth must last the rest of the elephant's life [8,11] (**Fig 14E**). | |
| **Spine** | N° of vertebrae: cervical: 7; thoracic: 18–21; lumbar: 3–5; sacral: 3–5; coccygeal: 18–34 [3] Tight intervertebral joints→limited spine flexibility. | |
| **N° of ribs** | 21 | 19-21 [11] |
| **Limbs/Feet** | Long limb bones with cancellous bone in place of medullary cavities [8,11]. Forearms, shanks and feet are fixed in pronation position. Circular feet with "cushion pads" of soft tissue [13], up to 5 toenails, soft soles (**Fig 16F**). Elephant feet have a variable skeletal feet anatomy [14] with 5 digits (phalangeal bones) and a sesamoid bone acting as a sixth digit (weight distribution) [15] | |
| **Skin** | Up to 2.5 cm thick, yet very sensitive to moisture-loss, sunburns and insect bites (requires regular mud baths). Elephants have two pectoral mammary glands [8,11]. | |
| **Gastrointestinal system** | Monogastric hindgut fermentation system. Total bowel length up to 35 m (**Fig 17F**). Most of the ingesta pass undigested. Liver has no gall bladder [8,11]. | |
| **Urogenital system** | Multipyramidal smooth kidneys (**Fig 19G**) and small urinary bladder. Testes are intraabdominal (caudal of the kidneys). Bipartite uterus (**Fig 19D**). Vulva with a well-developed clitoris is located between the hind legs [8,11] (**Fig 2D**). | |
| **Cardiovascular system** | Heart (weight 12–21 kg) displays a double-pointed apex [8,11], (**Fig 18D**). | |

*(Continued)*

**Table 2.** (Continued)

| Parameter | Asian elephant (*Elaphas maximus*, EM) | African elephant (*Loxodonta africana*, LA) |
|---|---|---|
| Respiratory system | Connective tissue in place of a pleural cavity (the thick visceral pleura is adhered to the parietal pleura). Lungs are attached to the diaphragm (breathing relies mainly on movement of the diaphragm) [8,11], (**Fig 7D**). | |

protocol below. The basic information presented here is intended as a brief summary of those aspects of elephant anatomy and physiology, which are immediately relevant for the necropsy process. More detailed information can be found in the pertinent literature and elsewhere [1,2,8].

## 3  Common and specific diseases of elephants under human care

Among the reported causes of death/reasons for euthanasia and necropsy findings of adult elephants under human care in western industrialized countries, the following diagnoses (non-infectious diseases) are frequently present [16] and should therefore be referred to in the anamnestic process and given attention to during the postmortem (PM):

- **Osteoarthritis/degenerative joint disease** [8,17].

- **Pododermatitis** [8,14,17] and **skeletal pathology** [14].

- **Renal disease** [16].

- **Tooth-related lesions** [8] (fractured teeth, alveolitis, sinusitis).

- **Trauma/fractures and inability to rise or move** [16].

- Aged female elephants are often diagnosed with diseases of the reproductive tract, including **cystic endometrial hyperplasia** [16], chronic **endometritis** [8,16,18], **leiomyomas** of the uterus [19] and **ovary cysts** [20], or **parturition complications** [16].

- **Aortic (abdominal) aneurysms** [8].

   Common and elephant-specific infectious diseases and zoonoses ($^Z$) repeatedly reported in elephants under human care include the following:

- **Tuberculosis**$^Z$ **(TB)** (*Mycobacterium tuberculosis, Mycobacterium bovis*, atypical mycobacteria) [21,22] and its differentials (*Corynebacterium pseudotuberculosis* infection, granulomatous pneumonia/lymphadenitis of other origin).

- **Salmonellosis**$^Z$ [23,24].

- **Elephant herpes** (*proboscivirus*, at least 5 different virus species) [25–33], manifesting as lymphofollicular vulvovaginitis (LA), papillomas, systemic hemorrhages (Asian elephant > African Elephant), and pulmonary lymphoepithelial nodules (LA). Differential diagnoses include *encephalomyocarditis virus (EMC-virus)*, *orbivirus,* salmonellosis or other bacterial septicemia, and vitamin E deficiency).

- **Encephalomyocarditis** (*EMC-virus*): Broad host range; acute fatal infections in LA with effusion, hydrothorax, ascites, pulmonary edema and congestion, and meningeal congestion; EM: Inapparent infections [34–38].

- Particularly wild elephants are also reported to be susceptible to **foot and mouth disease**$^Z$ (*FMD-picornavirus*) [39–41] and **anthrax**$^Z$ (*Bacillus anthracis*)] [42].

Infections with *Mycobacterium tuberculosis* and *Mycobacterium bovis* should always be ruled out during the necropsy and tested for appropriately. Due to the zoonotic potential of Mycobacteria species and its considerable incidence in elephants, which live in close relations to humans or are kept under human care, the sequence of dissection steps during the PM (**Section 6.4**) and adequate infection prophylaxis (**Section 4**) must be observed, especially in cases with unknown or suspected TB-status (also observe applicable local legal regulations for targeted handling with TB-suspected/TB-confirmed biological samples).

## 4 Workplace safety/infection prophylaxis and personnel protective equipment (PPE)

Apart from common workplace safety and infection prophylaxis precautions during standard PM examinations [43], elephant necropsies imply additional safety risks and potential zoonotic risks, which must be considered (**Section 3**). All persons involved in dissection of the elephant body, sampling and processing of organ and tissue specimens, disposal and transport of condemned animal materials and waste, and cleaning and disinfection of materials, instruments and facilities, must be informed about the potential infection and workplace safety risks and be instructed about the applicable precaution measures in advance of the necropsy. Furthermore, qualified first-aiders must be nominated prior to the necropsy (and present during the necropsy), the completeness of the first-aid kits and the operability of first aid equipment (*e.g.,* eye showers) have to be checked and the correct emergency numbers and contact addresses must be clearly indicated on the telephone. Diseased, immunosuppressed, underaged, and pregnant persons must not participate in the necropsy. In addition to the general safety instructions provided here, relevant safety notes/pictograms are also indicated in the respective protocols below. S1 Fig (in **Supporting information**) comprises all pictograms used in figures (**Figs 1-20**) and **Supporting Information** files, such as protocols and handouts.

   **All personnel** with contact to potentially infectious elephant tissues must constantly wear

- **adequate respiratory and eye protection** (*i.e.,* NIOSH/FDA-approved N95, N99, or N100 masks) combined with safety goggles and/or face shields (standard surgical masks are not adequate to protect from *Mycobacterium tuberculosis* infections) – in cases of suspected or confirmed TB, respiratory full-face masks of higher protection levels, such as powered air-purifying respirators (PAPRs) with sufficient battery packs are recommendable, or mandatory, depending on the local legal regulations on targeted handling with TB-suspected/TB-confirmed biological samples;

- (double) latex/nitrile **gloves** or chemical resistant rubber gloves;

- protective clothing such as disposable, water-repellent **coveralls with hoods and overshoes** (Tyvek gowns). The margins of the gloves and the openings of the overshoe/rubber boot shafts should be masked/taped with duct tape (**Fig 5C**);

   and should **not use private cellphones** or similar devices to take pictures during the necropsy.
   **Persons involved in the gross dissection** of the elephant body should also

- wear **rubber boots** with slip-resistant soles and robust (rubber) **aprons** and.

- **cut-resistant gloves** (if appropriate), and

- **keep proper distance to persons using chainsaws, axes, or cleavers**.

   **Persons operating engine-powered reciprocal- or chainsaws** or using axes/cleavers must

- **be familiar with/trained in the operation of the instruments.**

- wear adequate **cut-protective apparel** (trousers, shoes) and **ear protection** (if applicable).

- wear additional **face-shields for splinter protection**.

- avoid or reduce **aerosol formation** and exposure to saw-dust aerosols during sawing of bones by using adequate tools (axe/cleaver < hand (pit) saw < reciprocal saw < band/circular saw < chainsaw) and available extractor hoods (rather saw outdoors than inside closed rooms).

   To **avoid accidents in conjunction with moving/lifting of (heavy) elephant body parts**,

- **ropes or chain slings** with adequate load bearing capacity should be used to fasten and lift limbs or the head, rather than clipping the crane hook into a skin incision (**Figs 1D** and **4E.**) which might rip open.

- **place body parts on the ground or on a platform** to work on them rather than work on "hanging" body parts (if so, double secure the body part from accidentally falling off the crane hook).

- **do not work under lifted or moving body parts** (**Fig 5A**).

- **do not crawl inside body cavities** (*e.g.,* to dissect the diaphragm).

   When sampling organ/tissue specimens, adhere to the common safety precautions for working with **chemical substances** (fixatives), **liquid nitrogen** and **dry ice** (gloves, goggles, adequate ventilation) [43].

## 5  Prearrangements

### 5.1  Necropsy site

The facilities used for elephant necropsy must meet the legislative regulations for animal necropsies (also considering the known/suspected TB status of the animal). Prior to an elephant dissection, accessibility of the access route and driveway, sufficient size and height of the unloading site, in-house transport ways (unloading site – necropsy room) and the necropsy room should be controlled, as well as the load-bearing capacity of the present crane ropes, chains and lashing straps. Knowledge of these conditions is most important for the planning of the necropsy process, *i.e.,* to decidecidethe elephant corpse can be transported into the necropsy room as a whole, or if dissection of the body into smaller transportable pieces is necessary. The water supplies should be equipped with sufficiently long hoses. Finally, adequate space/room (lockable, with cooling) must be available for storage of the carcass/animal waste (approximately 2,500–5,000 kg, respectively 5–8 m$^3$ in case of an adult elephant) until it is collected by a certified rendering plant. The present work does explicitly not provide any recommendations for "field-necropsies" of elephants, since the (minimal) legal requirements regarding the provided work and infection protection measures and the necropsy/animal waste and sewage disposal may substantially vary in different countries.

### 5.2  Materials and equipment

An elephant necropsy can generally be performed using standard large animal necropsy instruments. Here, a sufficient number of large sharp necropsy knives (at least two per person) and knife sharpeners are necessary. The availability of additional, "heavy duty" necropsy instruments, such as axes, large hammers and chisels, and engine-powered saws (reciprocal saws or chainsaws) with appropriate protective gear (see above) is beneficial. For transport of heavy elephant body parts and necropsy disposal, shovels or large dustpans, big plastic tubs, and wheelbarrow(s) or carts on rollers are convenient. For collection, fixation and storage of organ/tissue samples, large containers, such as plastic buckets with tight fitting lids are useful. Fixatives, sample containers, leak proof styrofoam boxes, water ice, dry ice and liquid nitrogen, plastic bags, labels, and supplies, as well as extra/replacement masks, PAPR batteries, disposable PPEs, pencils and writing boards/clipboards, and approved tuberculocidal disinfectant should be present in sufficient quantities. For documentation of findings, sufficiently long tape measures and folding rulers, as well as a photo camera with charged extra batteries and appropriate illumination should be present. A complete checklist of suggested necropsy instruments, equipment and supplies is provided in S1 Table.

### 5.3 Personnel

An elephant necropsy generally requires a well-coordinated collaboration of 8–16 persons. Ideally, the necropsy is led by one experienced pathologist in charge who is responsible for coordination, execution, and control of all aspects of the necropsy. For an efficient and smooth necropsy and sampling workflow it is beneficial to assign the available personnel into separate necropsy groups/teams of 2–4 persons with defined tasks, as outlined in S2 Table. Here, the number and size of the different teams and their assigned tasks depend on the number of available pathologists and helpers, on the sequence of the necropsy steps (dissection sampling documentation), as well as on the available space and the scheduled extent of examined organs and collected organ/tissue specimens, which have to be planned and considered in advance of the necropsy. The scheduled operational sequence/workflow of the necropsy and sampling procedure, and the correspondingly defined duties of all involved necropsy teams and team members (including their assigned workspaces, instruments and sampling supplies that have to be prepared) must be communicated to all involved persons by the leading pathologist long enough before the beginning of the necropsy. Ideally, each team should have one member experienced in large animal necropsy and sampling. S2 File provides a structured checklist for the pathologist in charge to plan necessary preparations and organizational arrangements before, during and after an elephant necropsy. The **Supporting information material** (S1-S3 Files) also provides detailed work- and sampling instructions for the different necropsy teams/staff. These instructions can be adapted to the requirements of the individual necropsy, printed, and provided to the necropsy teams/staff members as "handout"-leaflet(s).

### 5.4 Dissection notes/documentation of findings and organ/tissue sample lists

A (copy-)template elephant necropsy form and an organ/tissue sample checklist are provided in the S1 File. These clearly structured forms guide through the entire necropsy and sampling process and are freely adaptable and expandable to the specific requirements of a given elephant necropsy.

### 5.5 Samples for superordinate (inter)national elephant studies

Elephant organ/tissue samples are permanently urgently sought-for by different international ongoing surveys and studies. Prior to the planned necropsy, the referring zoo-veterinarian(s) and the pathologist(s) should always contact the European Association of Zoos and Aquaria (EAZA) Ex situ Programmes (EEP) Coordinator and/or Veterinary Advisor to ask for required samples and current address of corresponding recipients and institutions. Collect appropriate tissue samples accordingly, if possible.

## 6 Postmortem procedure

The sequence of all relevant processes during an elephant necropsy is illustrated and described below, along with specific technical tips for dissection/dismemberment of elephant body parts and organ systems. Please note the indicated workplace-safety and infection-protection hints. The practical execution of all necropsy steps is also described step-by-step in the work-instruction leaflets for the different necropsy teams provided in the S2 File.

### 6.1 Clinical history

As with all necropsy cases, all relevant information related to the case should be enquired in advance of the necropsy (as detailed in the elephant necropsy-gross examination form provided in S1 File). These should comprise all essential identification documents and records of the medical history of the animal and its herd, including the (known/suspected) TB-status.

### 6.2 Storage, transport and unloading of an elephant body

To minimize the extent of autolysis, the transport of the elephant body to the necropsy site and the PM should take place as soon as possible after the elephant's death and be organized accordingly (refer to the checklist for the planning/

organization of an elephant necropsy, provided in the S2 File). The engaged transport company (or fire department/ technical relief service) must be informed about the accessibility of the pick-up and the unloading place/necropsy site (road widths, one-way streets, U-turn options, weight/length restrictions, *etc.*). Furthermore, transport conditions have to guarantee the prevention of leaking body fluids and a visual cover. Especially in warm weather conditions, or/and, if longer transport ways apply, the elephant body/abdomen should be refrigerated, *e.g.,* using large amounts of ice bags/blocks. Upon arrival at the necropsy site, the elephant body is unloaded (dump truck/crane) and moved to the necropsy room, depending on the in-plant facilities/premises (S2 File). Involved personnel should already wear appropriate PPE (including respiratory protection). Since the elephant body is dismembered in lateral recumbency, it is advantageous to position and transport the body in a way that any (anamnestically recorded) lesions/injuries are present at the upper body side. For workplace-safety during transport of the elephant body on a crane, it is most important to cinch/secure the elephant body appropriately, using stable lashing belts/chains (observe maximum load limits), rather than cambrels/meat hooks placed into skin incisions (if so, stay away from the danger zone, as long as the hook carries load, see **Fig 1**). The lifting height should be limited to the absolutely necessary extent. All persons must keep an appropriate distance and never stand below the suspended load.

### 6.3 External examination

As with any other large animal necropsy, the external examination (as detailed in the elephant necropsy-gross examination form provided in S1 File) involves documentation of identifying markers (chips, tattoos, distinguishing features), confirmation of death and estimation of autolysis degree, assessment of the physical condition and nutritional status, examination of skin, feet, and body orifices (eyes, ears, mouth, trunk (nostrils), anus, vulva, prepuce and penis), mammary gland (**Fig 2**), and superficial lymph nodes. If the technical prerequisites for determination of the elephant body weight (truck container scale or crane scale) are not present, the body weight (W) may roughly be estimated from the body length (L: base of the forehead to the base of the tail), and the chest girth (G), according to the formula provided by Sreekumar and Nirmalan [45]:

$$\textbf{W [kg]} = \textbf{--1010} + \textbf{0.036} * (\textbf{L [cm]} \times \textbf{G [cm]}).$$

All lesions are photographed and sampled, as appropriate. Particular attention should be paid to the occurrence of lymph node lesions suspicious of TB-infection[c], such as granulomatous/caseous lymphadenitis ± calcification (if

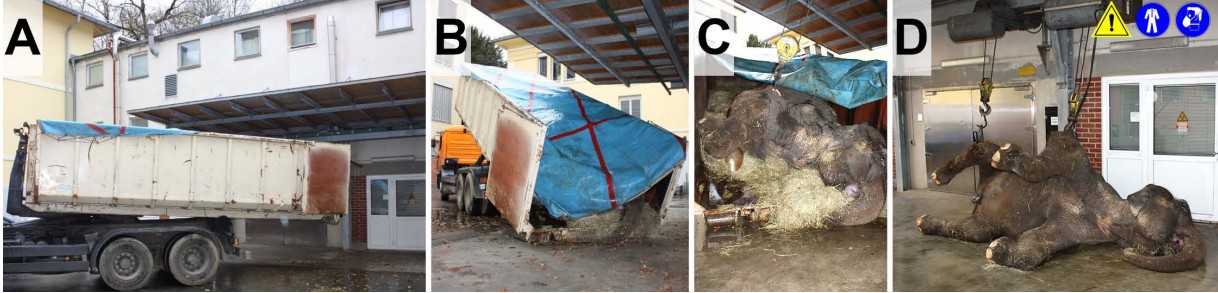

**Fig 1. Unloading (A, B) and crane transport (C, D) of elephant body (~3.5 t) to the necropsy room.** Here, large cambrels/meat hooks inserted in skin incisions are used to lift the body (the use of stable lashing belts/chains should be preferred). During crane transport persons must wear appropriate PPE, keep distance, and never stand below suspended load (⚠ 🛈 🛈).

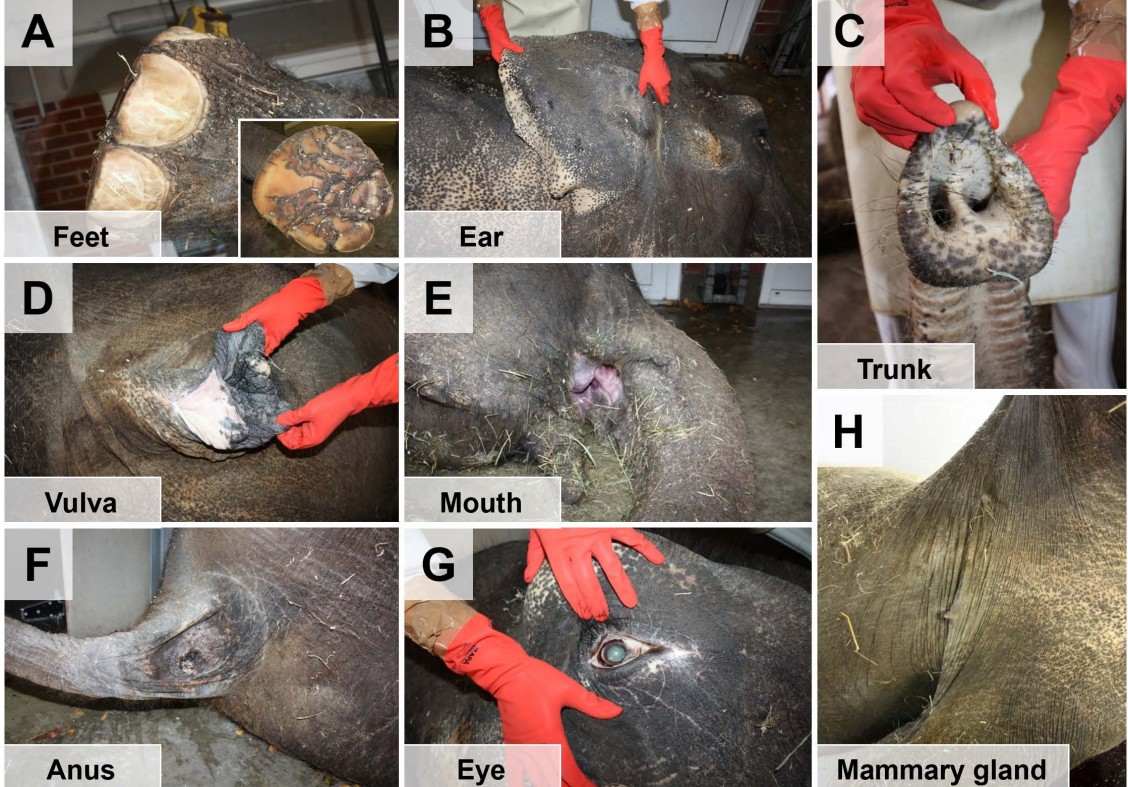

**Fig 2. External examination of an aged female Indian elephant.** Feet (**A**), ears (**B**), trunk (**C**, note the presence of a single -two in African elephants- "finger-like" extension dorsal of the nostrils), vulva (**D**, note the large distance to the anus), mouth (**E**), anus (**F**), eyes (**G**) and mammary gland (**H**, located between the front legs).

appropriate, switch to higher classes of respiratory protection). The present guideline does explicitly not schedule the application of acid-fast staining kits to determine the presence or absence of mycobacteria to rule out TB during the ongoing necropsy, because the numbers of mycobacteria may be very low in TB granulomas and the presence of mycobacteria is vice versa not pathognomonic for TB.

### 6.4 Dismemberment of the elephant body

For workplace safety and an efficient necropsy process, the elephant body is dismembered at a suitable location by a few, experienced staff members (here referred to as team "Dismemberment", see **Section 5.3**). Removed organs/body parts are then moved away from the dismemberment site and examined by separate necropsy teams (corresponding to organ systems or body parts, see **Section 5.3**). The teams further dissect the organs/tissues, detect and describe all pathological alterations and arrange their proper photo documentation and sampling (as specified in the work-instruction leaflets for the different necropsy teams, provided in the S1 **and** S2 Files). Here, the dismemberment process is shown and described in the sequence of factual events (**Figs 3**–**7**). In **Fig 3** the dismemberment is schematically summarized and outlined, details are shown in **Figs 4**–**7**. The processing, examination and sampling of different organ systems is subsequently described in **Sections 6.5.1**–**6.5.6** (**Figs 8**–**20**) and the **Supporting information material** (S1 and S2 Files).

- For dismemberment and allocation of body parts/organ systems to the different necropsy teams (**Sections 6.5.1**–**6.5.6**), the **elephant body is placed in lateral recumbency**, considering any anamnestically present lesions (upper side).

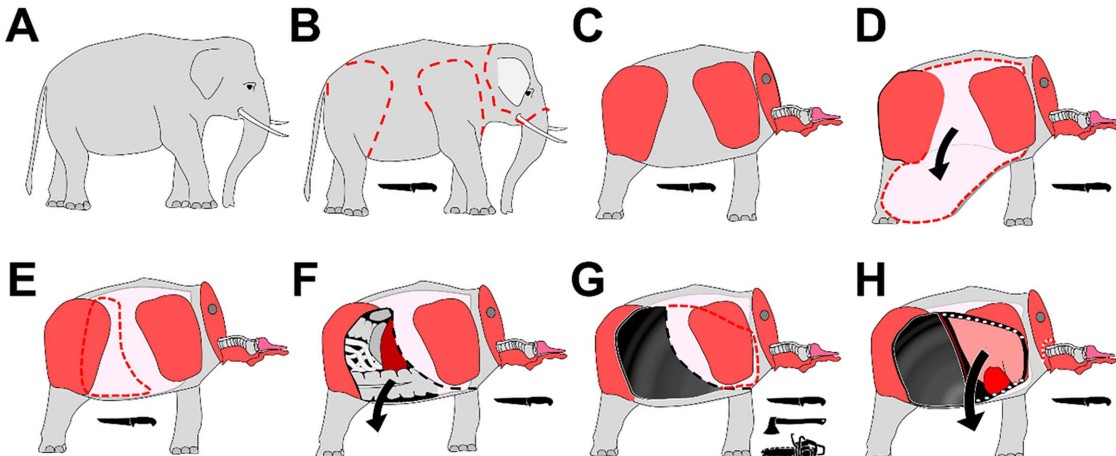

**Fig 3. Schematic illustration of the sequence of dismemberment of the elephant body.** Used necropsy instruments are indicated by pictograms. Section lines are indicated by *red dotted lines*. **(A)** The body is placed in lateral recumbency. **(B, C)** Removal of upper limbs and of the head (tongue and neck organs remain connected to the body). **(D)** Removal of the skin from the lateral abdomen. **(E)** The abdominal wall is removed. **(F)** Removal of abdominal and pelvic organs. **(G)** Removal of the lateral thoracic wall. **(H)** Removal of the thoracic and neck organs. Afterwards, the carcass is turned over, and the contralateral limbs are removed.

- First, **tongue and neck organs are dissected/mobilized** (approached from the ventral aspect of the lower jaw) and the **head is removed**, disarticulating the atlanto-occipital joint (**Fig 4**). For this step, it is advantageous to cinch and slightly lift the body of the elephant. Now, the mandibular, and the superficial and profound cervical lymph nodes can be examined. The separated head is fixed, secured, and moved from the dismemberment site for further examination (**Section 6.5.1**).

- Then the **upper fore- and hind limbs are removed**. For this purpose, the limbs are appropriately clinched/secured and slightly lifted. The front leg is removed together with the scapula; the hind leg is disarticulated in the coxofemoral joint (**Fig 5**). For easier transport and examination, it is advantageous to remove both feet separately (detach in the carpal/tarsal joints and label feet appropriately). The separated limbs are fixed, secured, and moved from the dismemberment site for further examination (**Section 6.5.2**).

- Then the **mammary gland** is removed and passed on for further examination **Section 6.5.5**).

- To access the **abdominal organs**, the skin is removed from the upper lateral abdominal body side, using knives and cambrels/meat hooks, and the **lateral abdominal wall is removed**, as shown in **Fig 6D**. Upon opening of the abdominal cavity, the intestines will usually dwell out the incision (postmortem gas accumulation). The abdominal cavity is examined for effusions, peritoneal alterations, and the position of the organs. If appropriate, samples are taken for microbiological examinations. When the abdominal wall is removed, large parts of the intestinal convolute will pour out on the ground. The spleen (if accessed from the left side) is removed. Then the stomach is removed continuously with the intestines. To facilitate this, the elephant body can slightly be lifted. The intestinal convolute is briefly examined in place and then moved from the dismemberment site for further examination (**Section 6.5.3**). Afterwards, the liver can be removed (**Section 6.5.3**).

- The **thoracic organs** are accessed by removal of the lateral thoracic wall[d] (**Fig 7B**). During opening of the thorax and examination of thoracic organs, all involved personnel must wear appropriate respiratory and eye protection (TB risk),

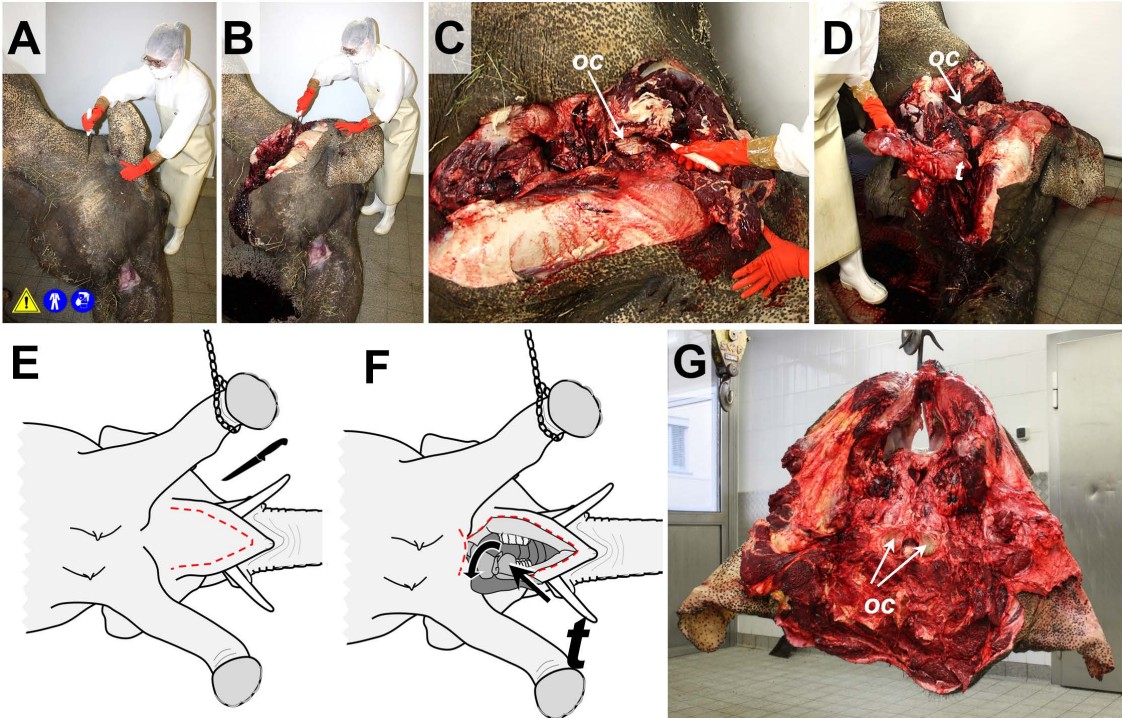

**Fig 4. Removal of the head. (A-D)** The head is disarticulated in the atlanto-occipital joint. The occipital condyles (*oc*) are indicated. **(D-F)** Tongue (*t*), pharynx and neck organs (larynx, trachea, esophagus, adjacent nerves and vessels, the thyroid and parathyroid glands, and cervical lymph nodes) are removed from the head (but left attached to the rump), as schematically illustrated in **(E-F)**: the *red dotted line* indicates the incision line at the ventral side of the lower jaw. Skin and mouth base are cut through following the medial contours of the lower jaw. The tongue and the adjacent neck organs are mobilized, severing the dorsal and lateral pharynx walls, and flipped backwards. **(G)** Caudal aspect of the removed head (hung up at the lower jaw). Note the occipital condyles (*arrows*) and removed neck organs.

while all other personnel should additionally keep appropriate distance (**Section 4**). First, the skin is removed from the thoracic wall. Then the ribs are cut through dorsally at the spine and ventrally at their insertion at the sternum. For this step, the use of an engine-driven reciprocal saw, or a chainsaw, is advantageous, although this implies a higher risk of aerosol production and requires special training and safety equipment (**Section 4**). Using a hand pit saw, axe and hatchet works as well but is more time-consuming and exhausting. The thoracic wall is folded ventrally (**Fig 7C**), while the connective tissue between the lungs and the parietal side of the thoracic wall is detached (**Fig 7E**). To facilitate this, the elephant body can slightly be lifted. After mobilization of the lungs and the heart inside the pericardium, the thoracic organs are briefly examined in place. Then the thoracic organs are removed from the thorax in continuity with the adjacent neck organs. The pluck is then moved away from the dismemberment site for further examination (**Section 6.5.4**). For workplace safety reasons, this guideline does not recommend to access and remove the thoracic organs from the diaphragm, since the pathologist performing the necropsy would have to step inside the abdominal cavity (depending on the size of the elephant and the pathologist). For infection prophylaxis, it is also recommendable to treat every elephant as if it were infected with TB, i.e., to always wear appropriate PPE.

- The carcass is then turned over and the contralateral limbs are removed (not shown). Subsequently, the organs of the urogenital system, the great abdominal blood vessels, deep inguinal lymph nodes, and the adrenal glands are excised the same way as in other large animal species (**Section 6.5.5**, **Fig 19A**).

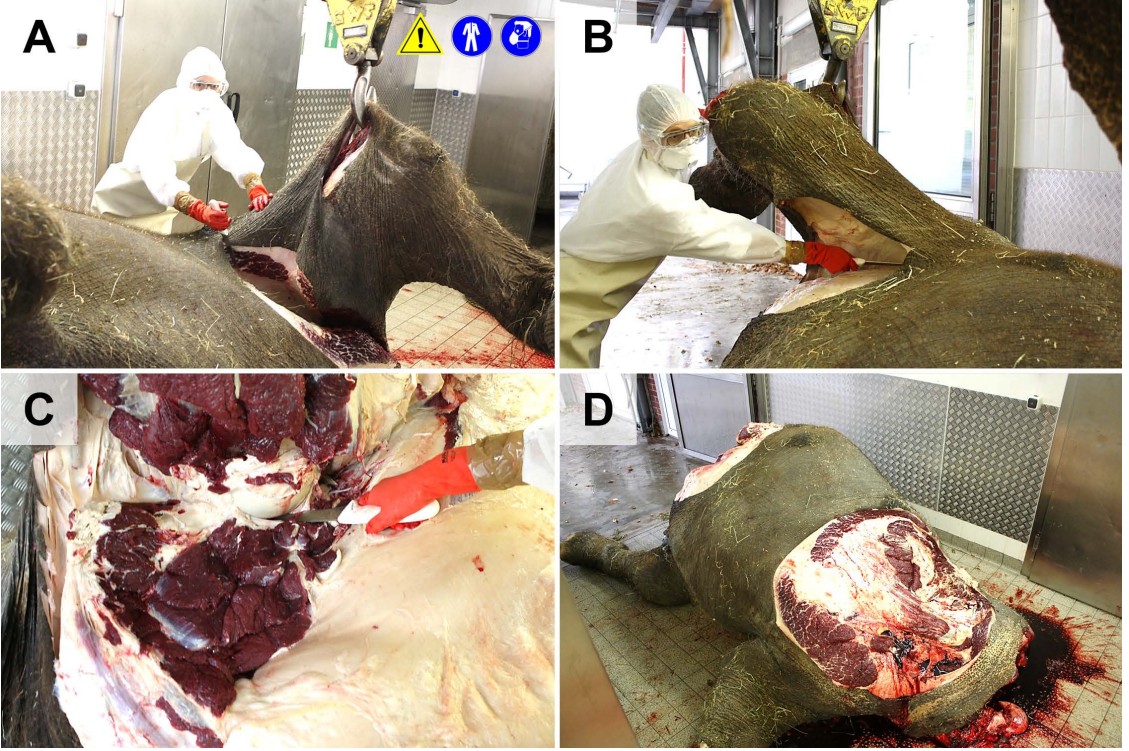

**Fig 5. Removal of the forelimb with the scapula (A) and the hindlimb (B) after disarticulation of the coxofemoral joint (C). (D)** Elephant body after removal of the right limbs.

- If appropriate, the muscles of the back, and the vertebral column/spinal cord are dissected subsequently (**Section 6.5.2**).

## 6.5 Dissection of distinct organs/tissues

The following sections describe the dissection of different organ systems and body parts, with special emphasis on elephant-specific anatomical peculiarities that require special necropsy techniques.

### 6.5.1 Dissection of the head.

- First, the **eyes** are removed and examined, as shown in **Fig 8**. For fixation for subsequent histological examination, modified Davidson's Fixative is recommended [44,46] (S1 Table).

- Then, the **trunk and the ear-pinnae** can be removed from the head. For examination of the trunk, the right and left nasal meatus are longitudinally opened from the nostrils to the proximal end of the trunk.

- The **lower jaw is removed** from the head after dissection and severance of the temporomandibular joint, as shown in **Fig 10D**. The oral cavity, teeth, the pharynx, and masticatory muscles are examined. The remaining head is further skinned (**Fig 10I**). If interested, the **vomeronasal organ** (Jacobson's organ) is examined and excised (**Figs 9**, **10**).

- The **brain** is accessed from the dorsal side, after removal of the mighty skullcap, which contains a thick central layer (40–60 cm in adult animals) of honeycomb-like structured, thin, brittle bone lamellae building up by the frontal sinus

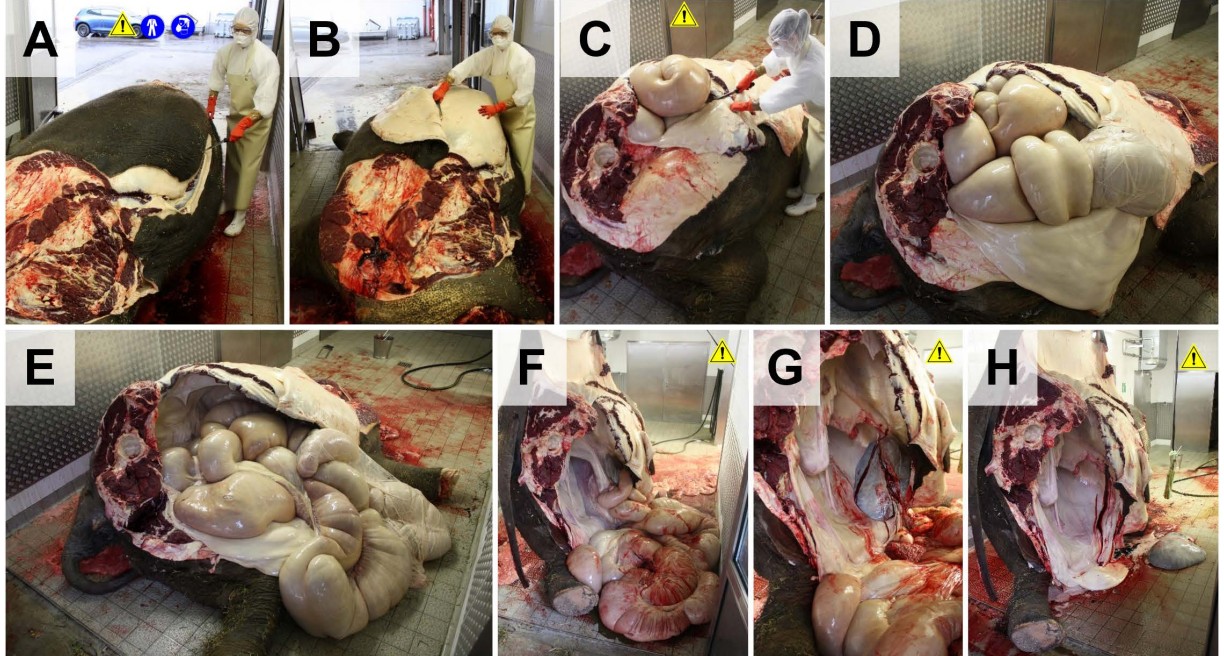

**Fig 6. Evisceration of abdominal organs. (A, B)** The skin is removed from the lateral abdomen. (**C-E**). Removal of the lateral abdominal wall. Note that intestines may contain large amounts of gas under pressure. **(F)** For removal of the intestines, the body may be lifted (workplace safety). **(G, H)** Intestines, stomach, spleen, and liver are removed. The urogenital organs, adrenals and large abdominal vessels are removed subsequently, or after removal of the contralateral hind limb.

cavity. When chopped with an axe or hatchet, these bone combs form sharp-edged splinters, which can chip away and fly off with high velocity, causing a considerable danger for eye/hand injuries (use appropriate eye protection/face shields). Alternatively, engine-powered saws can be used to access the cranial cavity (use appropriate protective gear, see S1 Table). **Figs 11** and **12** illustrate the access to the brain schematically and in photo-images. For the explanatory video see S1 Video.

- After removal of the skullcap, the meninges are examined and severed before the brain is removed from the cranial cavity (**Fig 12I**). Due to the prominent temporal lobes of the elephant brain, the opening in the skullcap must be wide enough to prevent damaging of the brain during its extraction. For fixation of the brain in 4% neutrally buffered form- aldehyde solution, the organ should be initially (incompletely) sectioned according to the LMU-Guide to Systematic Dissection of an Elephant Brain (S3 File and S2 Video), to allow sufficient tissue penetration by the fixative solution. The correct proportion of tissue: fixative solution of approximately 1:10, *i.e.,* ~35–40 l (!) must be kept in mind and the fixative solution should be renewed after 6–12 hours. A detailed step-by-step protocol for post-fixation trimming of the brain (regarding specific anatomical peculiarities of the elephant brain) for subsequent histopathological examination is provided in the S3 File and the corresponding S2 Video.

- After removal of the brain, the cranial fossae can be examined and the pituitary gland, cranial nerve roots and the tri- geminal ganglia can be excised, as illustrated in **Fig 12M**.

- **Examination of the external, middle, and inner ear**. After removal of the pinna, the long (30–50 cm in adult elephants) external ear canal is dissected from its outer orifice to the tympanic membrane (**Fig 13C**), using hammer and chisel to

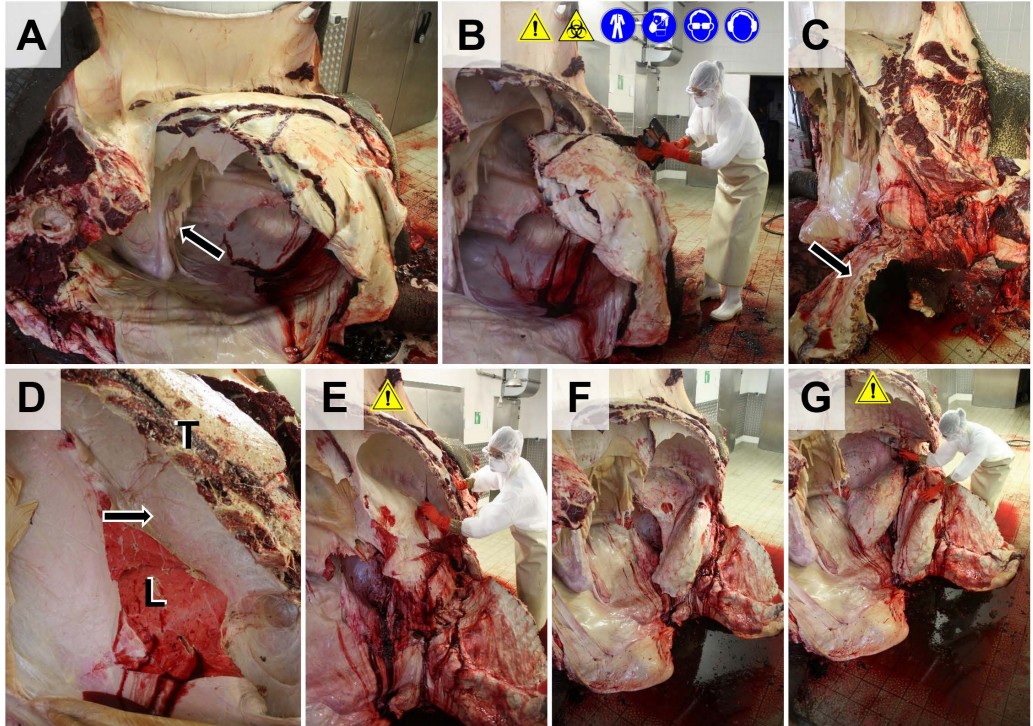

**Fig 7. Evisceration of thoracic organs. (A)** Caudal aspect of the eviscerated abdominal cavity (here, the urogenital organs still remain in the abdominal cavity, *arrow*). The skin is removed from the lateral thoracic wall. **(B)** Removal of the lateral thoracic wall. Here, a chainsaw is used for transection of the ribs. **C.** Ventrally folded thoracic wall (*arrow*). **(D-F)** Dissection of the connective tissue between lung (*L*) and thoracic wall (*T*). For removal of the thoracic organs, the body may be lifted (workplace safety). **(F)** Caudal view into the opened thorax. The right lung is mobilized. **(G)** Mobilization of the left lung. After mobilization of both lungs, the mediastinum, and the heart, the complete pluck (thoracic organs plus neck organs and tongue) is removed.

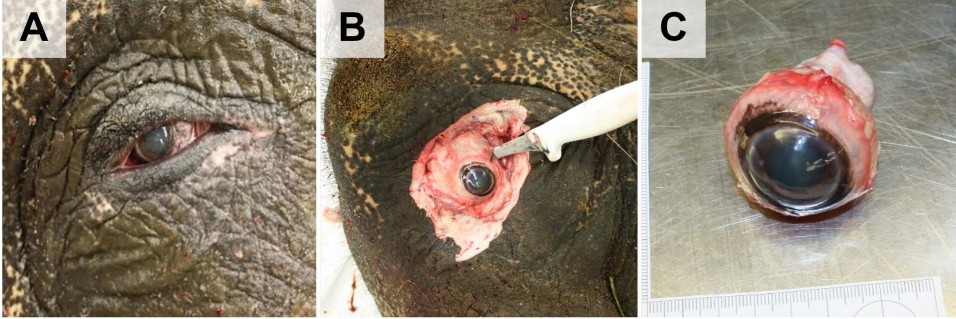

**Fig 8. Eye dissection. (A)** (Left) eye. **(B)** Excision of the ocular globe. **(C)** Extracted ocular globe.

remove the surrounding temporal bone. The external ear canals are frequently completely filled with cerumen and/or purulent exudate (see S1 Video). The tympanic cavity is comparably narrow and difficult to locate, since a tympanic bulla is not present in elephants, and the bone surrounding the tympanum is also widely permeated by pneumatized sinuses/ cavities. For preparation of the middle and inner ear, a triangular piece of bone containing the petrosal part of the temporal bone, the tympanum and the proximal part of the external ear canal is excised from the skull, as demonstrated in **Fig 13E**-**13I** and further dissected/processed, as appropriate (depending on the type of scheduled downstream analysis,

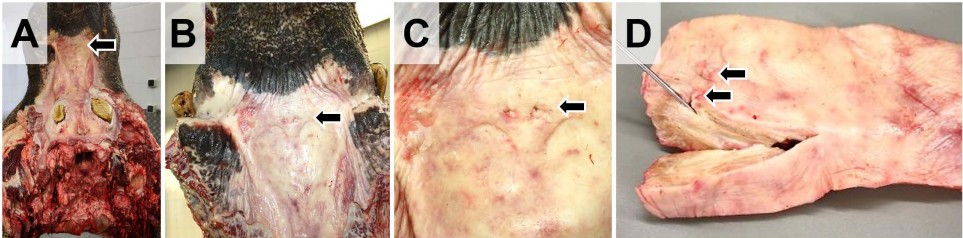

**Fig 9. Dissection of Jacobson's organ. (A-C)** The bilateral openings of the vomeronasal organ are visible on the surface of the rostral oral mucosa of the maxillary bone (*arrows*). **(D)** Excised tissue sample from the roof of the mouth. *Arrows* mark the oral openings of the Jacobson's organ. One canal of the organ is longitudinally opened and inserted with a probe.

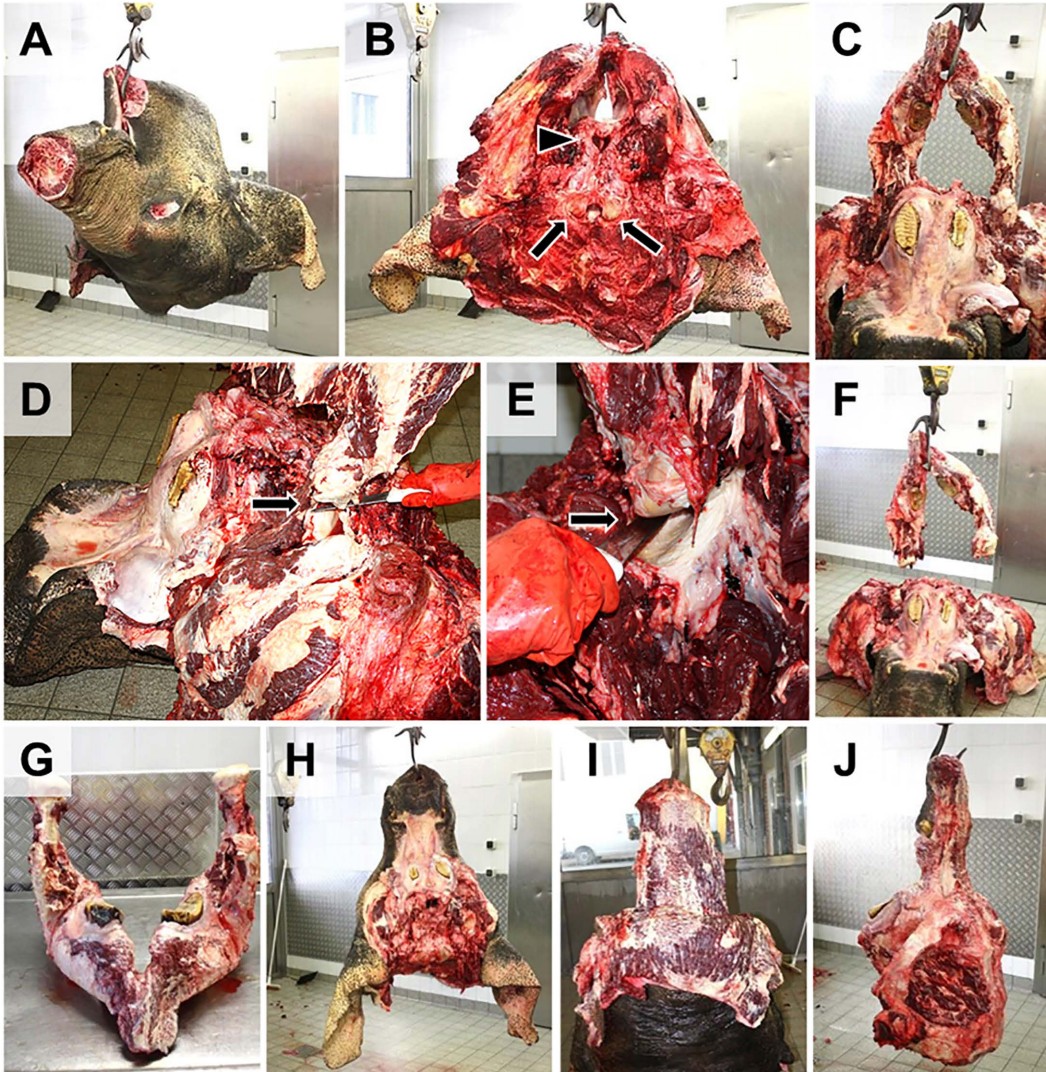

**Fig 10. Dissection of the head (part one). (A)** Head after removal of eyes and trunk. **(B)** Caudo-ventral aspect of the head. *Arrows* mark the condyles. The *black arrowhead* indicates the section profile of the ventral nasal meatus. **(C-E)** Removal of the lower jaw. The soft tissue of the cheeks is removed from the jaws **(C)** to access **(D)** and sever **(E)** the temporomandibular joint (*black arrow*). **(F, G)** Removed lower jaw. **(H)** Ventral aspect of the head after removal of the lower jaw. **(I, J)** Removal of skin and musculature from the head.

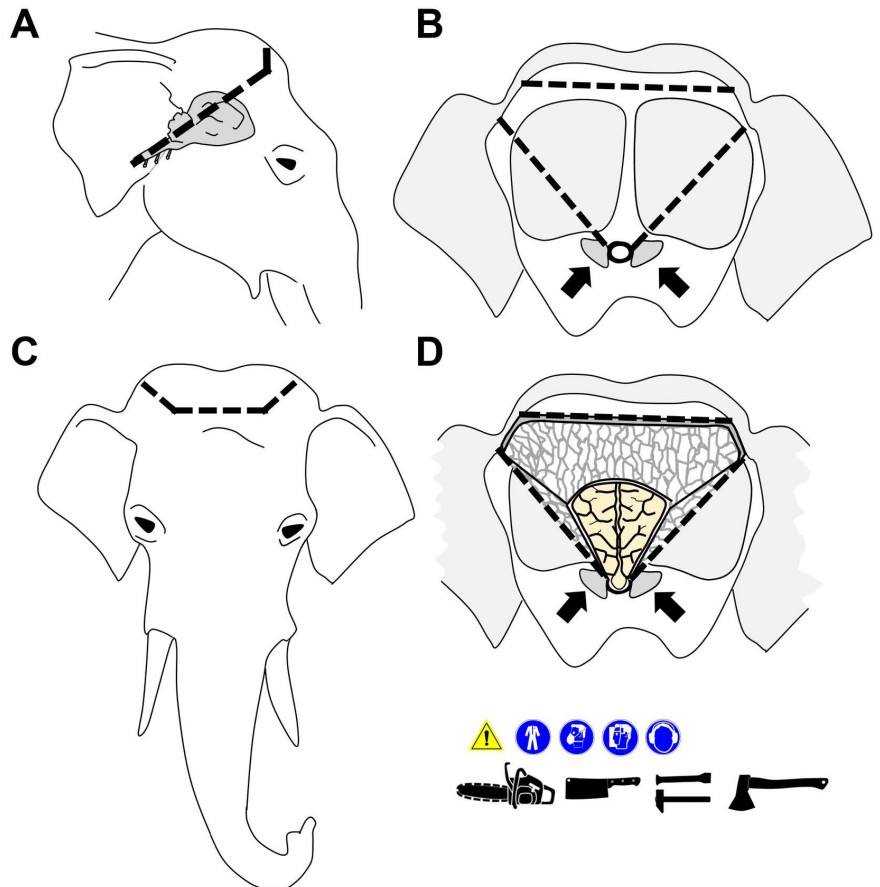

**Fig 11. Schematic illustration of the removal of the skullcap to access the brain.** Cutting lines are indicated by *black dotted lines*. *Arrows* mark the condyles as landmarks for orientation. Used necropsy instruments and applicable workplace safety provisions are indicated by pictograms **(A)** Lateral view. The relative position of the brain is indicated. **(B)** Caudal aspect. **(C)** Frontal aspect. **(D)**. Caudal aspect of the skull after removal of the skullcap. Note the thick layer of honeycomb-like, thin, and splintery bone forming the frontal sinus.

*e.g.,* Computer tomography (CT)/Magnetic resonance imaging (MRI), Scanning electron microscopy (SEM)/Transmission electron microscopy (TEM), **Fig 13K**). *Note that decalcification of the (massive) petrosal bone for histological examination of the middle/inner ear may require a very long time (several weeks to months) and probably impair the samples histomorphology. For histological and ultrastructural analyses of inner ear structures, the membranous labyrinth (cochlea, semicircular duct ampulla, etc.) is therefore better directly (micro)dissected from the petrosal bone under a stereomicroscope as previously described [47] and then fixed and processed separately (no decalcification required).*

- **Teeth lesions** and associated pathological conditions, such as alveolar abscesses or sinusitis are comparably frequent in elephants under human care (**Fig 12R**). Teeth are examined for abrasion, decay, fractures, and gingival lesions. The dental alveoli can be opened from latero-dorsal (upper jaw) or from the ventro-lateral side (lower jaw) using hammer and chisel to examine the tooth roots. On this occasion, the number of remaining reserve teeth for replacement of abraded molars (6 in total, see **Table 1**) can be determined. Extraction of teeth, particularly of molars, is facilitated after cooking of excised jaw sections for 12–24 hours in >70°C water, as illustrated in **Fig 14**. Note that strict legal regulations apply for the possession and transfer of elephant body parts. Extracted tusks/ivory should therefore only be transferred to authorized persons or be disposed appropriately (document transfer and obtain signed/dated receipt, see **S1 File**).

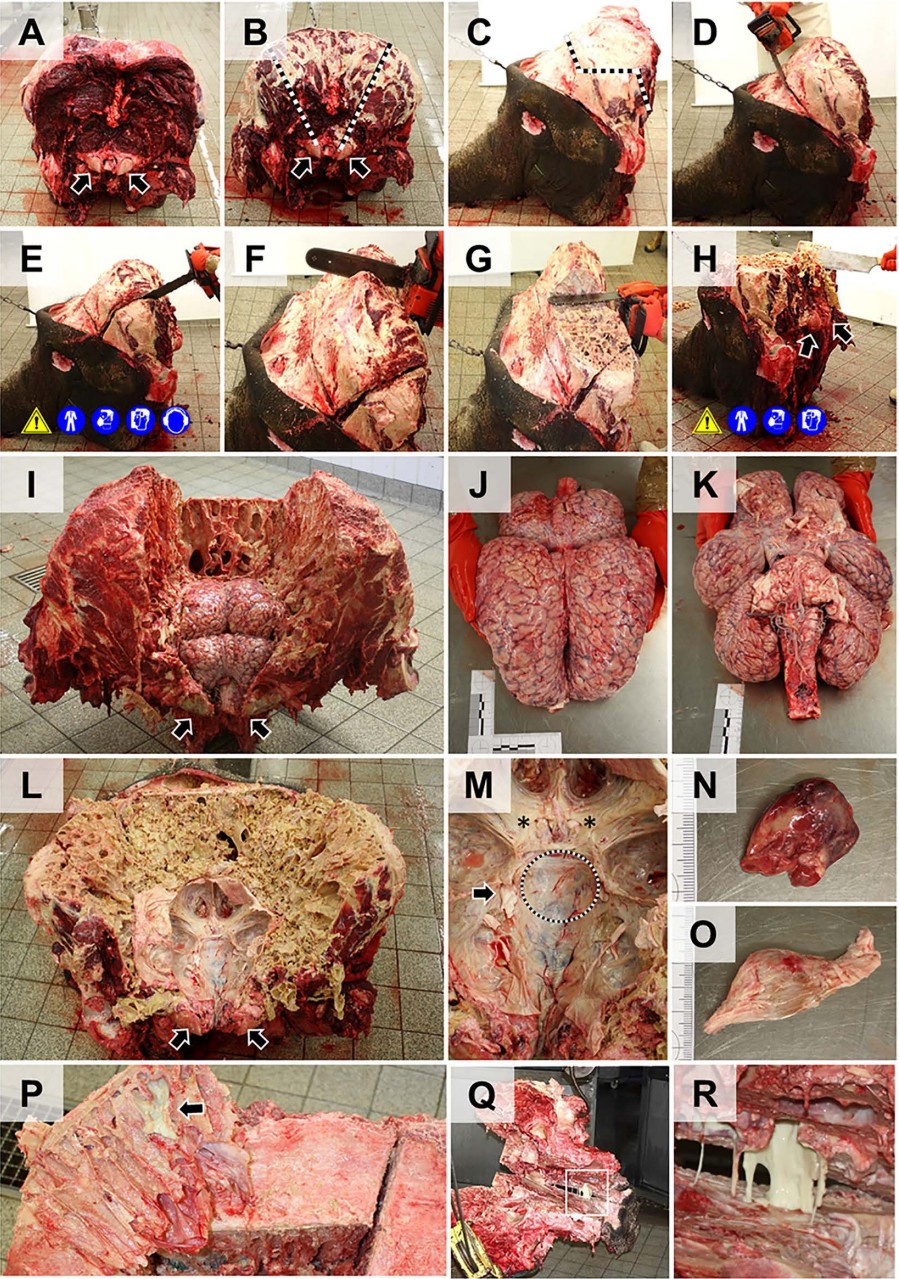

**Fig 12. Dissection of the head (part two): Removal of the brain (compare to Fig 11). (A)** Caudal aspect of the head. Note that in the present example, the trunk and the lower jaw have not been removed in advance. *Arrows* mark the condyles. **(B)** Removal of neck musculature attached to the occipital bone. **(C)** Lateral aspect. The *dotted lines* in **(B)** and **(C)** indicate the incision lines for removal of the skullcap. **(D-F)** Incision into the frontal sinus. Here a chainsaw is used (workplace safety). **(G)** Removal of a large portion of the dorsal skullcap by an additional horizontal incision. **(H)** Further removal of bone with a hatchet (workplace safety). **(I)** Caudal aspect of the brain surface after removal of the skullcap. **(J, K)** Removed brain (dorsal **(J)** and ventral **(K)** aspect). Ruler length = 10 cm in sections of 1 cm. **(L)** Caudal aspect of the head after removal of the brain. Note the large convexities for the prominent temporal lobes of the brain. **(M)** Dorsal aspect of the middle cranial fossa. *Asterisks* indicate optical nerves. The *arrow* marks the position of the (left) trigeminal ganglion. The *dotted line* indicates the incision line for extraction of the pituitary gland, which is covered by a thick layer of dura mater. **(N)** Excised pituitary gland. **(O)** Excised trigeminal ganglion. Ruler length = 10 cm in sections of 1 cm. **(P-R)** Frequently observed pathological findings include purulent sinusitis of the frontal sinus **(P)** and the maxillary sinus **(Q)**; **(R)**: detail enlargement of the *boxed area* in **(Q)**.

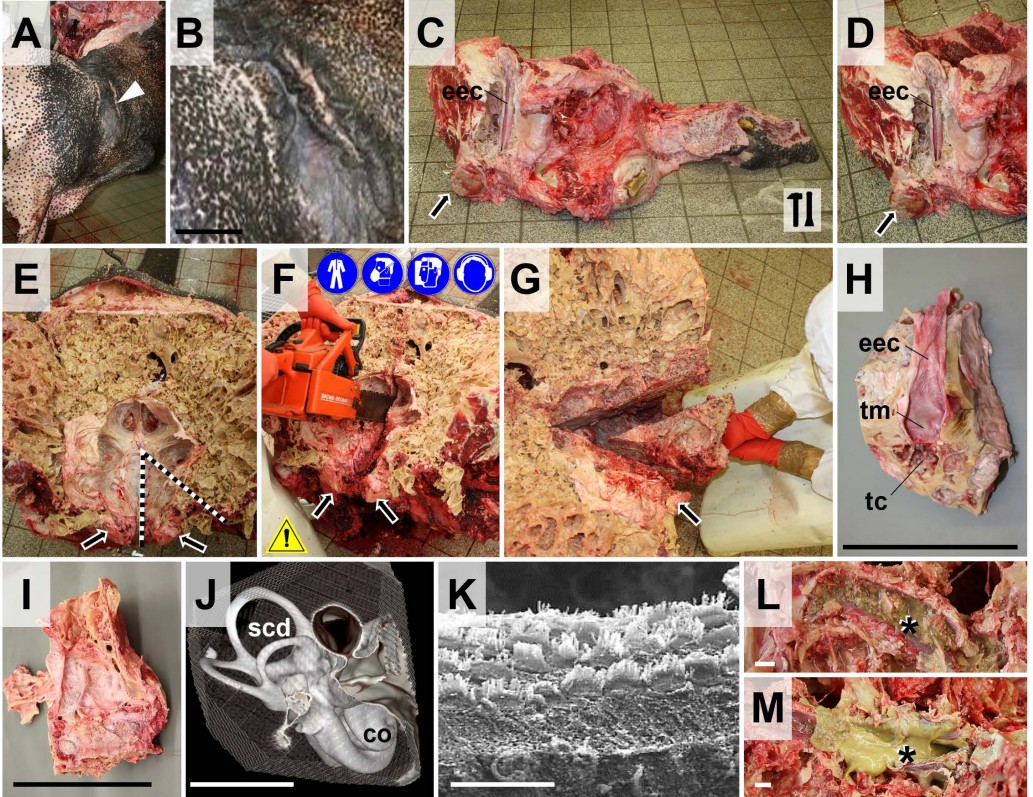

**Fig 13. Ear dissection. (A)** Pinna. The opening of the external ear canal (*eec*) is indicated by a *white arrowhead*. **(B)** Detail magnification of the opening of the external ear canal. **(C, D)** Longitudinally opened right external ear canal (*eec*). The *arrow* marks the condyle. Note that the external ear canal was cleaned for photography (frequently, the external ear canal is filled with cerumen and/or purulent exudate (see S1 Video). **(E-G)** Excision of the petrosal part of the temporal bone with the middle and inner ear. Note that the images in **(A-D)** and **(E-G)** derive from different elephants/cases. **(E)** Caudodorsal aspect of the skull after removal of the brain (compare to **Fig 12L**). The *dotted lines* indicate the incision lines for excision of the petrosal bone. **(F)** Excision of the petrosal bone. Here, a chainsaw is used (workplace safety). **(G)** Removal of a triangular, wedge-shaped sample containing the petrosal bone with the middle and inner ear. Lateral **(H)**, and medial **(I)** aspect of the excised petrosal bone (compare to **G**). The proximal part of the external ear canal *(eec)*, the tympanic membrane *(tm)*, and the middle ear (tympanic) cavity *(tc)* are indicated. Bars = 10 cm. **J.** 3D-image of the inner ear labyrinth with the cochlea *(co)* and semicircular ducts *(scd)* reconstructed from Computer Tomography (CT) images acquired in the sample shown in **(H)** and **(I)** (Siemens-Somatom AS helical 64-multislice CT scanner: pitch 0.55, 140 mA, 120 kVp, rotation time 1 sec., slice thickness 0.6 mm). Bar = 5 mm. **(K)** Scanning electron micrograph (Zeiss DSM 950) of the outer hair cells in the cochlea, imaged in a sample prepared from the tissue sample shown in **(H)** and **(J)**, demonstrating the good preservation of tissue morphology/ultrastructure. Bar = 20 μm. **(L, M)** Frequently observed pathological findings include otitis externa with excessive accumulation of cerumen in the external ear canal **(L)** and purulent exudate **(M)**. Bars = 1 cm.

- Before decocting, the head is mid-sagittally split and further divided to access and examine the nasopharyngeal passages and perinasal sinus (**Fig 12Q**).

### 6.5.2 Musculoskeletal system, spinal cord, peripheral nerves, and feet.

- **Skeletal muscles**, **bones** and **bone marrow**, large **joints** of the fore and hind limbs, and large **peripheral nerves** are accessed, dissected, examined, and sampled as in other large animal species (**Fig 15A** and **15B**). Particular attention is paid to joint lesions indicative of osteoarthritis/degenerative joint disease and traumatic alterations.

- For dissection of distinct **spinal cord** segments, it is easier to dissect single vertebrae and access and remove the spinal cord directly from the vertebral canal, then to split the entire vertebral column, or to saw open single vertebrae (**Fig 15C** and **15D**).

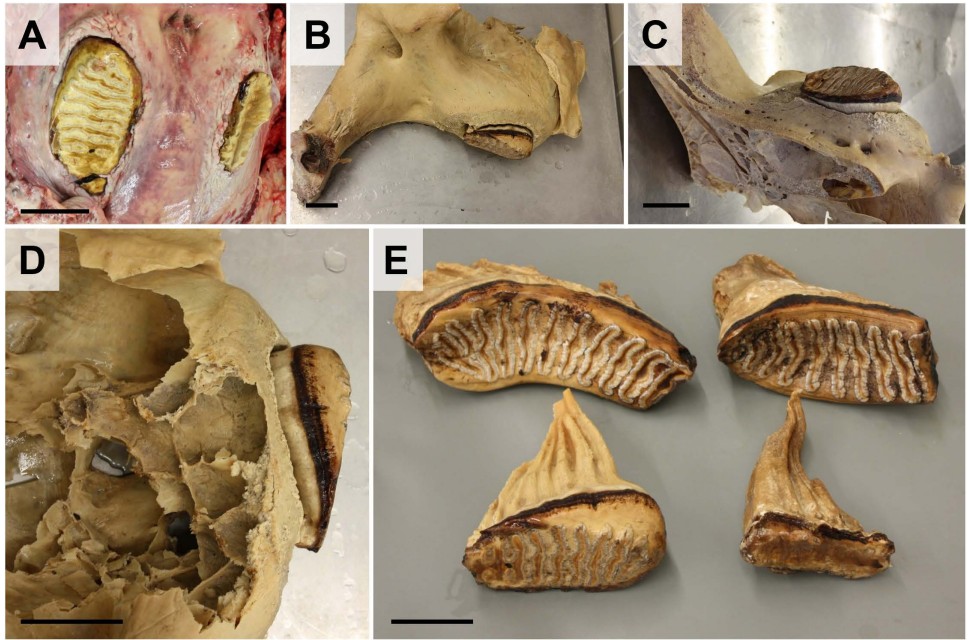

**Fig 14. Teeth dissection.** **(A)** Teeth are macroscopically inspected for lesions (tooth decay, fractures, abrasion, *etc.*). **(B-E)** Extraction of teeth is facilitated by decocting of the jawbone sections. **(B-D)** Decocted fragment of the upper jaw. Lateral **(B)** and medial **(C)** aspect. **(D)** Teeth are accessed from the alveolus and removed. **(E)** Extracted molar teeth of a 64-year-old female elephant. Upper row: Left and right molar teeth of the upper jaw. Lower row: Left and right (severely abraded) mandibular molar teeth. Bars = 10 cm.

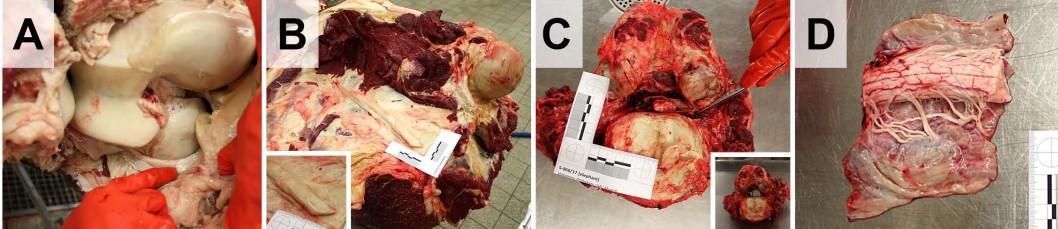

**Fig 15. Dissection of large joints of the limbs (A, here: knee joint), skeletal muscles and peripheral nerves (B, here: sciatic nerve), and spinal cord segments (C, D).** **(C, D)** Caudal aspect of the excised fourth cervical vertebra. The spinal cord is removed from the vertebral canal. See text for details. Ruler length = 10 cm in sections of 1 cm.

- Due to the frequent occurrence of pododermatitis, degenerative alterations in metacarpal/metatarsal, and phalangeal joints, and of lesions of tendons, synovial bursae, digital bones and sesamoid bones [8,14], a thorough examination of the **feet** is important (**Fig 16**). The feet are removed from the legs in the metacarpal/metatarsal joints and should instantly be labeled appropriately (front/hind/right/left), to warrant the correct allocation of findings, photo-images, and collected tissue samples. Application of imaging techniques, such as Computer tomography or Magnetic resonance imaging, are most useful to identify and locate skeletal foot lesions, and/or to confirm/follow up clinical findings/ tentative diagnoses prior to dissection of the feet [14]. The dissection of feet is exemplarily illustrated in **Fig 16C**–**16G**. A band saw (aerosol-formation – wear appropriate respiratory protection) can be used to produce (multiple parallel) sagittal sections through the foot for examination of all relevant anatomical structures.

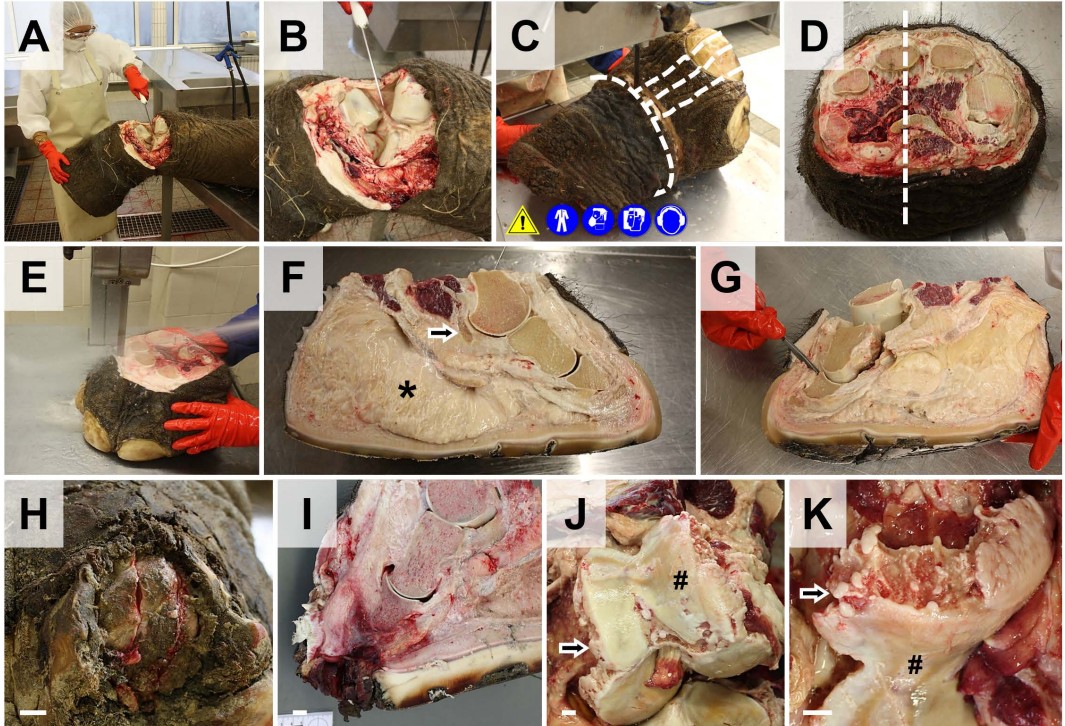

**Fig 16. Dissection of feet. (A, B)** The foot is severed in the metacarpal joint. **(C-E)** For examination of all relevant anatomical structures, the foot is sectioned in parallel sagittal slices, using a band saw. Incision lines are indicated by *white dotted lines*. **(D)** Dorsal aspect of the left hind foot transversally sectioned at the level of the metatarsal bones. The *dotted line* corresponds to the orientation of the section in **(E-G)**. **(F)** Mid-sagittal section through the foot at the level of the third digit. The *arrow* marks a sesamoid bone. The *asterisk* indicates a large "cushion" of connective tissue. **(G)** Dissection and examination of toe joints. **(H-K)** Frequently observed pathological findings include pododermatitis **(H, I)** and osteoarthritis in aged animals **(J, K)**. **(J, K)** Severe osteoarthritic lesions of the knee joint of a 67-year-old animal with loss of articular cartilage, eburnation (#) and osteophyte formation (*arrows*) at the femoral condyles **(J)** and the patella **(K)**. Bars = 1 cm.

### 6.5.3 Spleen and gastrointestinal system.

- Apart from the great length of the intestines (~30 m) and their heavy weight due to the intestinal contents, the dissection of the elephant gastrointestinal tract is principally performed as in other monogastric large animal species (**Fig 17E**–**17J**). After removal of the spleen, the liver and the pancreas from the intestinal convolute, the mesentery, its lymph nodes, and the omentum are removed and examined. The intestines are separated, displayed in rows, and opened up longitudinally. The ingesta/feces are examined and removed before examination and sampling of the mucosal surfaces. For examination and sampling of the parenchyma of spleen, liver and pancreas, the organs are serially sliced into parallel slabs of a few cm thicknesses. Esophagus, pharynx, and tongue (attached to the pluck) are separated from the thoracic organs and examined after inspection of the cervical lymph nodes, the larynx, and the thyroid/parathyroid glands (**Section 6.5.4** and **6.5.6**).

### 6.5.4 Respiratory and cardiovascular system.

- Trunk, nasal passages, and perinasal sinus are examined during dissection of the head (**Section 6.5.1**).

- After examination and removal/sampling of the larynx, cervical lymph nodes, blood vessels and nerves, and the thyroid/parathyroid glands (**Section 6.5.6**), the esophagus, pharynx, and tongue (attached to the pluck) are separated from the thoracic organs and examined (**Section 6.5.3**).

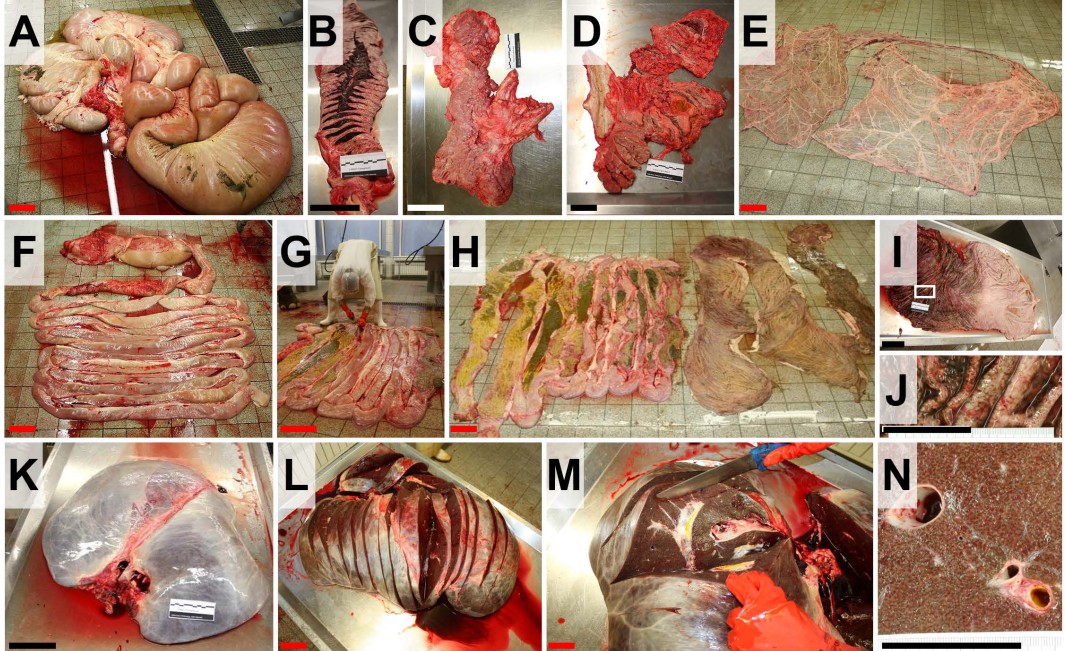

**Fig 17. Dissection of the spleen and the gastro-intestinal tract. (A)** Intestinal convolute after removal from the abdominal cavity. Bar = 15 cm. **(B)** Spleen, serially sliced for examination of parenchyma. **(C)** Excised pancreas with adjacent vessels. **(D)** Serially sliced pancreas. Bars in B, C and D = 10 cm. **(E)** Omentum and mesentery spread out for gross examination. Bar = 15 cm. **(F-H)** After removal of the mesentery, intestines are displayed in rows **(F)**, longitudinally opened **(G)**, and examined **(H)** after removal of the ingesta/feces. Note the huge caecum (hindgut fermentation). Bars = 15 cm. **(I)** Opened stomach (Bar = 10 cm) with hyperemic/congested gastric mucosa **(J**, Bar = 5 cm). **(K-N)** Dissection of the liver. The liver **(K)** is serially sliced **(L)** for gross examination **(M)**. Bars = 10 cm. **(N)** Closer view in the section surface of the liver parenchyma, demonstrating a distinct lobular pattern. Bar = 5 cm.

- Mediastinum, thymus (if present), mediastinal and tracheobronchial lymph nodes are examined and sampled (with particular attention paid to alterations suspicious of TB infections, such as proliferative/granulomatous lymphadenitis with soft caseation/cavitation/calcification). Larynx, trachea, bronchi, and large pulmonary vessels are longitudinally opened. To examine the lung parenchyma, the tissue is serially sectioned into parallel (frontal) slices using a large sharp knife (**Fig 18B**).

- The heart is removed from the pericardium and dissected as in other large animal species (**Fig 18D–18F**). The great arteries, including the abdominal aorta, are longitudinally opened, and thoroughly examined for the presence of degenerative vascular alterations and aneurysms.

### 6.5.5. Urogenital system.

- Dissection of the urogenital tract is performed, as in other large animal species (**Fig 19**). The kidneys, the adrenal glands, and in male elephants the (intraabdominal) testes, epididymides, and spermatic ducts, are removed from the abdominal cavity and standardly examined (**Figs 19H** and **20A**).

- In male elephants, the urinary bladder and the proximal part of the urethra and the adjacent accessory sexual glands are severed and removed from the pelvic cavity. Penis and prepuce are separated from the ventro-caudal abdominal wall and examined separately (not shown).

- The female genital tract (ovaries, salpinx, uterus, cervix, vagina, vulva, and clitoris) is completely removed via the pelvic outlet (the pelvic floor remains intact) by a circular incision around the vulva (**Fig 19C**).

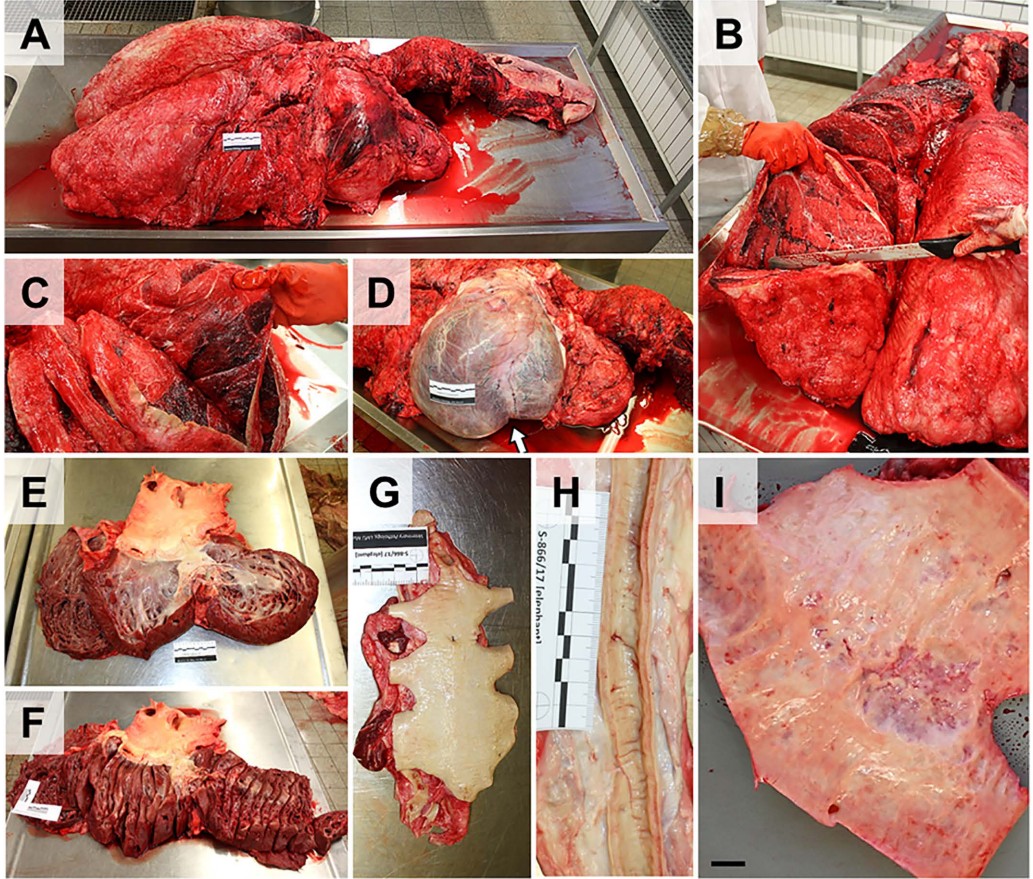

**Fig 18. Dissection of lungs, heart and great vessels. (A)** Pluck (tongue, neck organs, lung, mediastinum and heart) after removal from the thorax (compare to **Fig 7G**). **(B)** The lung parenchyma is serially sliced for gross examination. Note the tarnished surface of the lungs (connective tissue of the fused visceral and parietal pleura). **(C)** Congestion and consolidation of the ventral part of the cranial lung lobe. **(D)** Heart after incision of the pericardium. Note the double-pointed apex of the heart (*arrow*). **(E)** Ventricles and atria are dissected as in any other large animal species. **(F)** The myocardium is serially sliced for gross examination. **(G, H)** Longitudinally opened thoracic **(G)** and abdominal **(H)** aorta. Ruler length = 10 cm in sections of 1 cm. **(I)** Frequently observed pathological findings in aged animals include atherosclerosis of the abdominal aorta. Bar = 1 cm.

- During examination of the ovaries and uterus, particular attention is paid to the presence of ovarian cysts, cystic endometrial hyperplasia, and leiomyomas, which frequently occur in (aged) elephant cows [8,16,19,20].

- If pregnant, the fetus (size, weight, position, and if applicable, complete necropsy), the placenta/embryonic membranes, and umbilical cord are examined and sampled. In cases of fatal birth complications, it might be necessary to saw the mother's pelvis open to extract the fetus (rather than dismembering the fetus).

### 6.5.6 Endocrine system.

- Dissection of the pituitary gland (**Section 6.5.1**), the (endocrine) pancreas (**Section 6.5.3**), and the gonads (**Section 6.5.5**) is described in the respective sections above.

- Thyroid glands (**Fig 20C**) are located bilaterally at the proximal trachea, ventrodistal of the larynx. The parathyroid glands are located directly adjacent to the thyroid or partially embedded in the thyroid parenchyma.

- The large adrenal glands (**Fig 20A**) are easily identified in the perirenal retroperitoneal adipose tissue (**Fig 19B**).

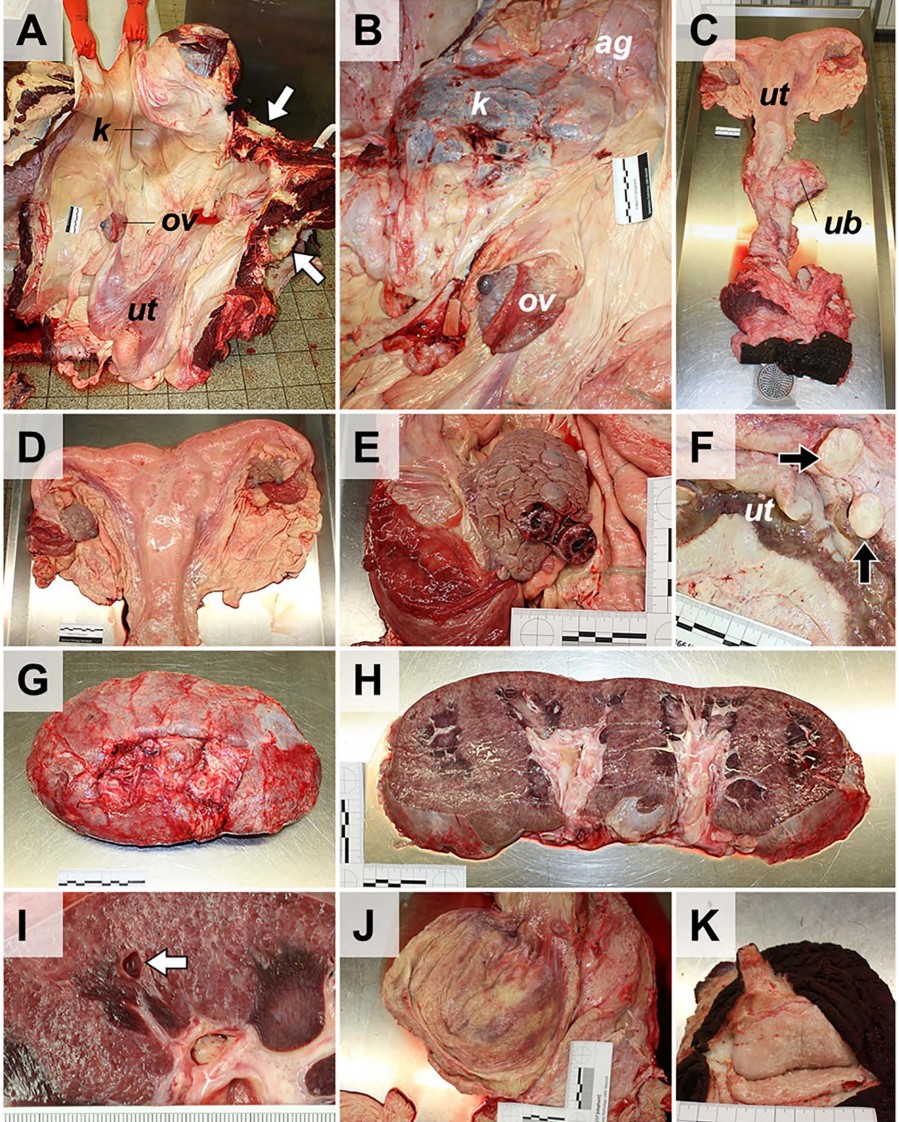

**Fig 19. Dissection of the urogenital tract of an aged female Asian elephant. (A)** Situs of the abdominal cavity after removal of the gastrointestinal organs and removal of the hind legs. *Arrows* mark the acetabuli, the pelvic floor remains intact. The position of the left kidney (*k*), the left ovary (*ov*) and the uterus (*ut*) are indicated. **(B)** Detail enlargement of **(A)** after removal of the peritoneal coverage of the kidney (*k*). The location of the adrenal gland (*ag*) is indicated. **(C, D)** Removed female genital tract. The comparably small urinary bladder (*ub*) is indicated in **(C)**. **(E)** Closer view of the left ovary displaying a hemorrhagic corpus luteum. **(F)** Longitudinally opened uterus with multiple nodular leiomyomas (*arrows*). **(G-I)** Dissection of the kidney(s). Excised **(G)** and sectioned **(H)** kidney. **(I)** Closer view of the renal cortex, with interstitial fibrosis and a small cyst (*arrow*). **(J)** Opened urinary bladder. **(K)** Sectioned mammary gland complex (inactive). Ruler length = 10 cm in sections of 1 cm.

## 6.6 Sampling of organ/tissue specimens

During the necropsy, organ/tissue samples are collected from pathological alterations discovered during macroscopic examination and corresponding to the predefined sample lists (S1 and S2 Files). Samples are collected and differentially processed, according to the scheduled downstream analysis methods, as outlined in S3 Table. To warrant a quick and sufficient penetration of tissue specimens sampled for histological analyses with fixative solutions, adequate volume

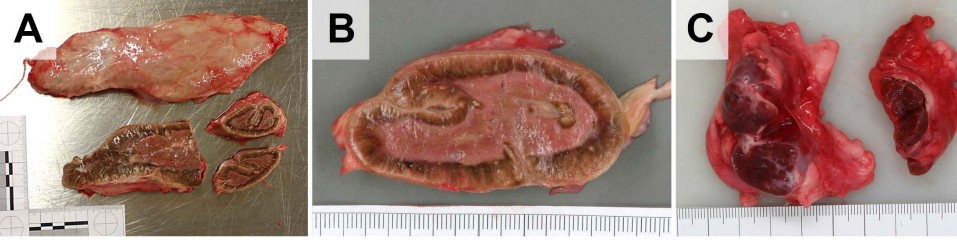

**Fig 20. Adrenal glands (A, B) and thyroid glands (C) from an aged female Asian elephant.**

proportions of tissue: fixative solution, renewal of the fixative solution in appropriate time intervals, and maximal sample/tissue slice thicknesses must be observed. For large organ specimens, it is recommended to exhaustively cut the tissue prior to fixation. In general, tissues should be cut into parallel slices of appropriate thickness (incomplete slicing for cohesion of single tissue slabs) prior to fixation, or after a short phase of immersion in the fixative solution. For the special case of brain dissection, follow the systematic dissection protocol in **Section 6.5.1** (S3 File and S2 Video). For an efficient sampling process during the necropsy, it is expedient if (scheduled) tissue samples are spaciously excided, placed on appropriately labeled disposable paper plates, and handed over to co-workers (sampling team) who subsequently take all necessary (sub)samples from the transferred tissue specimen and process them accordingly (refer to S3 Table and S1 File).

## 6.7  Disposal of the carcass and disinfection

Subsequent to the completion of the necropsy and sampling, the elephant carcass and remaining organs (approximately 2,500–6,000 kg, respectively 3–8 m³), and the generated necropsy waste (disposable PPE, supplies) and sewage must be disposed appropriately (corresponding to the local legal regulations). Until the animal waste is collected by a certified rendering plant, it must be stored in suitable waste containers/ rooms (lockable, ideally with refrigeration). Used non-disposable PPE and necropsy instruments, as well as the premises must be thoroughly cleaned and disinfected. For disinfection, only approved tuberculocidal disinfectants should be used according to the manufacturer's recommendations. The persons performing these work(s) must wear appropriate PPE. Traffic and possession of elephant tissues are regulated by strict national and international laws (*e.g.,* Convention on International Trade in Endangered Species of Wild Fauna and Flora, CITES). Therefore, ivory/tusks or other body parts of the elephant must not be passed to unauthorized persons. If such elephant parts are returned to the referring zoo or the former owner of the animal, the transfer should be precisely documented (photo, weight) and confirmed by signature and date.

## 6.8  Time requirements

The duration of an elephant necropsy depends on the scheduled extent of examined organs and collected organ/tissue specimens, of the number, experience, and discipline of the available personnel, and of the proper organization, planning, and preparation of the necropsy process. A complete necropsy of an adult elephant with >10 persons will usually take at least 4–6 hours.

## 6.9  Animals, ethics statement

The present work was based on seven routine elephant necropsy cases at the Institute of Veterinary Pathology at the Center for Clinical Veterinary Medicine, Ludwig-Maximilians-Universität München, Munich, Germany from 12/21/2016-10/12/2022. The age of these animals ranged from 46 to 67 years, their weight from 2800 to 5000 kg. The whole study

was performed in accordance with the relevant legal regulations and approved by the management at each participating institution, and where applicable, by zoo research committees.

All persons being pictured in figures/photographs have given written informed consent to appropriate image use.

## Discussion

For most veterinary pathologists, the necropsy of an elephant is a rare and often unexpected event. Performance of a successful, timely and complete elephant postmortem examination requires extensive prearrangements (normally with a short period between planning and execution), organization skills, distinct safety precautions, particular infrastructures, and special dissection techniques [3–6,43]. The protocols presented here were intended as a comprehensive yet concise guideline for veterinary pathologists being in charge of an elephant necropsy, warranting for the completeness of the dissection procedure and the sufficient quality and quantity of the sampled tissue specimens. Special emphasis was placed on a meaningful sequence of necropsy steps, personal infection prophylaxis and workplace safety, as well as on technical details of the dissection of such elephant organs/tissues that are difficult to access. Also rarely examined organs are considered, such as the vomeronasal organ or the inner ear. The provided checklists and work-instructions (S2 File) were designed to guide the "managing pathologist" and the different "necropsy teams" through the necropsy, to ensure the completeness of all scheduled examinations and sampling plans. The necropsy protocol(s) and checklists can be individually adapted to the given circumstances, *i.e.,* the existing infrastructures (location, transport vehicles, crane lifting capacity), and the numbers of available assistants, the time frame, and the number of required/requested organ/tissue samples, and/or special examinations, such as Computer tomography or Magnetic resonance imaging techniques [14].

Given the comparably high prevalence of tuberculosis in elephants under human care and the implied zoonotic risk [22,48,49], adequate and effective personal infection prophylaxis is essential during the necropsy and the processing of organ/tissue samples [43]. Moreover, the use of "heavy equipment" such as cranes and chainsaws to move and dissect heavy elephant body parts implies special risks, demanding for special workplace safety precautions. Therefore, the indicated safety notices (and applicable legal regulations) should always be strictly observed. In particular, the use of electric- or gasoline-powered chainsaws should be carefully considered. On the one hand, these tools efficiently simplify and accelerate the necropsy process. If operated by an adequately trained user wearing appropriate cut protection clothing, shoes, and a plexiglass splinter shield, the use of chainsaws is definitely not more dangerous than the use of axes, cleavers, and hand or wire saws, especially when used to remove the massive, brittle, and sharp-splintery skull/frontal sinus bone to access the brain. On the other hand, however, chainsaws (as well as bandsaws) produce large amounts of (potentially infectious) aerosols, especially in closed rooms [50]. In cases with suspicious lymph node alterations, the use of chainsaws should therefore carefully be reconsidered.

Owing to a sense of collegiality, and to take full advantage of the rare opportunity for collection of fresh elephant tissue samples for ongoing superordinate multicenter elephant studies, the referring zoo-veterinarian(s) and the pathologist(s) in charge should also always remember to contact the corresponding institutions/scientists (**Section 5.5**) prior to an elephant necropsy to consider their sample lists.

## Supporting information

**S1 Fig. Pictograms used in figures (Figs 1–20) and supporting information.**
(TIF)

**S1 File. Material.** Standard elephant necropsy form and organ tissue sampling lists.
(DOCX)

**S2 File. Material.** Checklist for the preparation and organization of an elephant necropsy with work instructions (handouts) for elephant necropsy teams/personnel.
(DOCX)

**S3 File. Material.** LMU-guide to systematic dissection of an elephant brain including table of S2 Video sections.
(DOCX)

**S1 Table. Materials and equipment checklist.**
(DOCX)

**S2 Table. Personnel, necropsy teams and tasks.**
(DOCX)

**S3 Table. Sampling and processing of tissue specimens.**
(DOCX)

**S1 Video. Extraction of the elephant brain.**
(MP4)

**S2 Video. LMU protocol for systematic dissection of an elephant brain.**
(MP4)

## Acknowledgments

The authors thank Lisa Pichl, Sandra Aumiller, Josef Grieser, Marold Handl, and Sebastian Hunger (LMU) for excellent technical assistance.

## Author contributions

**Conceptualization:** Almuth Falkenau, Andreas Blutke, Ninja Kolb, Alexandra Rieger, Isabelle Lutzmann, Katharina Erber, Clara Kaufhold, Lina Eddicks, Marco Rosati, Sonja Fiedler, Anna Gager, Effrosyni Michelakaki, Elena Dell'Era, Timo Lorenzen, Martin Zöllner, Andreas Brühschwein, Andrea Meyer-Lindenberg, Kaspar Matiasek, Monir Majzoub-Altweck, Julia Heckmann, Marco Roller, Lukas Reese, Barbara Lang, Markus Menzinger, Nicole Richter, Robert Fitz, Lukas Pfaudler, Christine Lendl, Hanspeter W. Steinmetz, Christine Gohl.

**Data curation:** Ninja Kolb, Alexandra Rieger, Isabelle Lutzmann, Katharina Erber, Clara Kaufhold, Lina Eddicks, Marco Rosati, Sonja Fiedler, Anna Gager, Effrosyni Michelakaki, Elena Dell'Era, Timo Lorenzen, Martin Zöllner, Andreas Brühschwein, Andrea Meyer-Lindenberg, Kaspar Matiasek, Monir Majzoub-Altweck.

**Formal analysis:** Ninja Kolb, Alexandra Rieger, Isabelle Lutzmann, Katharina Erber, Clara Kaufhold, Lina Eddicks, Marco Rosati, Sonja Fiedler, Anna Gager, Effrosyni Michelakaki, Elena Dell'Era, Timo Lorenzen, Martin Zöllner, Andreas Brühschwein, Andrea Meyer-Lindenberg, Kaspar Matiasek, Monir Majzoub-Altweck.

**Investigation:** Almuth Falkenau, Andreas Blutke, Ninja Kolb, Alexandra Rieger, Isabelle Lutzmann, Katharina Erber, Clara Kaufhold, Lina Eddicks, Marco Rosati, Sonja Fiedler, Anna Gager, Effrosyni Michelakaki, Elena Dell'Era, Timo Lorenzen, Martin Zöllner, Andreas Brühschwein, Andrea Meyer-Lindenberg, Kaspar Matiasek, Monir Majzoub-Altweck, Julia Heckmann, Marco Roller, Lukas Reese, Barbara Lang, Markus Menzinger, Nicole Richter, Robert Fitz, Lukas Pfaudler, Christine Lendl, Hanspeter W. Steinmetz, Christine Gohl.

**Methodology:** Almuth Falkenau, Andreas Blutke, Ninja Kolb, Alexandra Rieger, Isabelle Lutzmann, Katharina Erber, Clara Kaufhold, Lina Eddicks, Marco Rosati, Sonja Fiedler, Anna Gager, Effrosyni Michelakaki, Elena Dell'Era, Timo Lorenzen, Martin Zöllner, Andreas Brühschwein, Andrea Meyer-Lindenberg, Kaspar Matiasek, Monir Majzoub-Altweck.

**Project administration:** Almuth Falkenau, Andreas Blutke, Julia Heckmann, Marco Roller, Lukas Reese, Barbara Lang, Markus Menzinger, Nicole Richter, Robert Fitz, Lukas Pfaudler, Christine Lendl, Hanspeter W. Steinmetz, Christine Gohl.

**Supervision:** Almuth Falkenau, Andreas Blutke.

**Validation:** Almuth Falkenau, Andreas Blutke, Julia Heckmann, Marco Roller, Lukas Reese, Barbara Lang, Markus Menzinger, Nicole Richter, Robert Fitz, Lukas Pfaudler, Christine Lendl, Hanspeter W. Steinmetz, Christine Gohl.

**Visualization:** Almuth Falkenau, Andreas Blutke, Ninja Kolb, Alexandra Rieger, Isabelle Lutzmann, Katharina Erber, Clara Kaufhold, Lina Eddicks, Marco Rosati, Sonja Fiedler, Anna Gager, Effrosyni Michelakaki, Elena Dell'Era, Timo Lorenzen, Martin Zöllner, Andreas Brühschwein, Andrea Meyer-Lindenberg, Kaspar Matiasek, Monir Majzoub-Altweck, Julia Heckmann, Marco Roller, Lukas Reese, Barbara Lang, Markus Menzinger, Nicole Richter, Robert Fitz, Lukas Pfaudler, Christine Lendl, Hanspeter W. Steinmetz, Christine Gohl.

**Writing – original draft:** Almuth Falkenau, Andreas Blutke.

**Writing – review & editing:** Almuth Falkenau, Andreas Blutke, Ninja Kolb, Alexandra Rieger, Isabelle Lutzmann, Katharina Erber, Clara Kaufhold, Lina Eddicks, Marco Rosati, Sonja Fiedler, Anna Gager, Effrosyni Michelakaki, Elena Dell'Era, Timo Lorenzen, Martin Zöllner, Andreas Brühschwein, Andrea Meyer-Lindenberg, Kaspar Matiasek, Monir Majzoub-Altweck, Julia Heckmann, Marco Roller, Lukas Reese, Barbara Lang, Markus Menzinger, Nicole Richter, Robert Fitz, Lukas Pfaudler, Christine Lendl, Hanspeter W. Steinmetz, Christine Gohl.

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
