## [Decision Letter · Decision Letter 0]

6 Oct 2025

Dear Dr. Falkenau,

We look forward to receiving your revised manuscript.

Kind regards,

Pierre Roques, Ph.D.

Academic Editor

PLOS ONE

Journal Requirements:

**Additional Editor Comments:**

well done it was really a huge work in all the meaning of this word

Reviewers' comments:

Reviewer's Responses to Questions

**Comments to the Author**

1. Is the manuscript technically sound, and do the data support the conclusions?

Reviewer #1: Yes

Reviewer #2: Yes

2. Has the statistical analysis been performed appropriately and rigorously?

Reviewer #1: N/A

Reviewer #2: N/A

3. Have the authors made all data underlying the findings in their manuscript fully available?

Reviewer #1: Yes

Reviewer #2: Yes

4. Is the manuscript presented in an intelligible fashion and written in standard English?

Reviewer #1: Yes

Reviewer #2: Yes

Reviewer #1: Creating an elephant-specific necropsy manual is of utmost importance for the education and training of veterinarians. Given the anatomical and physiological uniqueness of these large mammals, and the scarcity of opportunities to perform necropsies on them, a detailed manual provides invaluable guidance. It not only allows practitioners to familiarize themselves with the complex arrangement of their organs and tissues, but also enables the standardization of procedures for sample collection and the identification of species-specific diseases. This will facilitate research into causes of mortality, the understanding of diseases that affect them, and ultimately contribute significantly to the conservation of these impressive animals, arming veterinarians with the knowledge necessary for accurate diagnoses and better management of their health

This manuscript presents technically sound scientific research, supported by a rigorous methodology and robust empirical data. The conclusions derived from this study are directly supported by the evidence collected, ensuring its reliability and validity.

Regarding statistical analysis, in this particular case, the data obtained were analyzed following the methods described and appropriate for descriptive studies, which allows for presenting clear and relevant findings without the need for more complex statistical tools.

The authors have made all the data supporting the findings described in the manuscript publicly available without any restrictions. The photographs included are self-explanatory, allowing for a clear understanding of the results, and the videos provided are of excellent quality, effectively complementing the information presented.

Given the size of the animal, an elephant necropsy inevitably requires the use of dangerous tools such as chainsaws or axes. This entails significant risks, not only due to the sharp edges left by the skull bones, but also due to the proximity of the operators when handling these tools. Therefore, while the authors mention the need for training and a suitable location, it is crucial to emphasize and reiterate the importance of biosecurity in this type of procedure, ensuring the safest possible working environment.

Reviewer #2: The authors of this article aim to provide a thorough overview of the necropsy process of elephants. It targets every veterinary practitioner or animal facility personnal facing elephant death. The authors not only provide a very well-illustrated, very complete guide on the necropsy process. They also provide a wide overview of the regulatory process in occidental countries, a wide overview about elephant pathologies (especially tuberculosis which is frequent in elephant and is one of the most common infectious disease in human causing death of around 1 millions people world-wide each year). The authors particularly insist on the critical role of safety in this large animal necropsy procedure due to the physical risks due to the size of the animal and the handling of potentially harmfull tools such as knives, chainsaws or hammers.

Even though the quality of the pictures in the main text could be slightly improved, the authors provide few videos to explain the process to remove the brain its dissection.

The supplemental material provide several checklists of the personnal and the material to have during the procedure to be fully operationnal. This aspect is well developped in the main text as the procedure is long, demanding in manpower and demands a well thought organisation.

The authors have provided complete , thourough guide which is easy to read and use as a stand-alone prcedure to follow.

The english is good and only one translation error could be observed on line 430 : "und" instead of "and"

**Do you want your identity to be public for this peer review?** For information about this choice, including consent withdrawal, please see our Privacy Policy

Reviewer #1: **Yes:** Francisco Pedraza-Ordoñez

Reviewer #2: **Yes:** Quentin PASCAL (DESV-AP, DVM)

---

## [Author Response · Author response to Decision Letter 1]

24 Nov 2025

Dear Academic Editor Pierre Roques and Reviewers Francisco Pedraza-Ordoñez and Quentin Pascal,

We would like to sincerely thank you for taking the time to review and provide feedback on our manuscript (Research Article) entitled “A pictural guide to postmortem examination of elephants” by Falkenau et al. We greatly appreciate your careful consideration of this comprehensive work.

We have addressed the minor revision points to the best of our ability. Specifically:

• A grammar and spelling check was conducted throughout the manuscript, including the correction in line 430 (“und” → “and”, now in line 433), also throughout the S1-3 Material.

• References in the main manuscript as well as references in S3 Material have been formatted according to PLOS ONE requirements.

• No substantive changes to the content were made.

Please find attached the tracked and untracked revised versions of the manuscript for your review.

• All figures were checked with PACE and quality adjustments were made to Fig 11 and S1_Fig.

We hope that these revisions meet with your satisfaction, and we look forward to your feedback.

Thank you once again for your time and consideration.

Sincerely,

Almuth Falkenau et al.

---

## [Editor Report · Decision Letter 1]

27 Nov 2025

A pictural guide to postmortem examination of elephants

PONE-D-25-31751R1

Dear Dr. Falkenau,

We’re pleased to inform you that your manuscript has been judged scientifically suitable for publication and will be formally accepted for publication once it meets all outstanding technical requirements.

Kind regards,

Pierre Roques, Ph.D.

Academic Editor

PLOS ONE

Additional Editor Comments (optional):

Thank you for this work that should be a reference for all veterinarians dealing with an elephant cadaver.
---

## [Editor Report · Acceptance letter]

PONE-D-25-31751R1

PLOS One

Dear Dr. Falkenau,

I'm pleased to inform you that your manuscript has been deemed suitable for publication in PLOS One. Congratulations! Your manuscript is now being handed over to our production team.

Kind regards,

on behalf of

Dr. Pierre Roques

Academic Editor

PLOS One